# Diagnosing Bottlenecks in Data Visualization Understanding by Vision-Language Models

## Abstract

Data visualizations are vital components of many scientific articles and news stories. Current vision-language models (VLMs) still struggle on basic data visualization understanding tasks, but the causes of failure remain unclear. Are VLM failures attributable to limitations in how visual information in the data visualization is encoded, how information is transferred between the vision and language modules, or how information is processed within the language module? We developed FUGU, a suite of data visualization understanding tasks, to precisely characterize potential sources of difficulty (e.g., extracting the position of data points, distances between them, and other summary statistics). We used FUGU to investigate three widely used VLMs (LLaMA-3.2, LLaVA-OneVision, and InternVL3). To diagnose the sources of errors produced by these models, we used activation patching and linear probes to trace information flow through models across a variety of prompting strategies. We found that some models fail to generate the coordinates of individual data points correctly, and these initial errors often lead to erroneous final responses. When these models are provided with the correct coordinates, performance improves substantially, suggesting that the downstream mathematical reasoning steps performed in the language module are sound. Moreover, even when the model generates an incorrect response, the correct coordinates can be successfully read out from the latent representations in the vision encoder, suggesting that the source of these errors lies in the vision-language handoff. We further found that while providing correct coordinates helps with tasks involving one or a small number of data points, it generally worsens performance for tasks that require extracting statistical relationships across many data points (e.g., correlations, clusters). Finetuning models on FUGU also fails to yield ceiling performance. These findings point to fundamental architectural constraints in current VLMs that might pose significant challenges for reliable data visualization understanding.[1]

## 1 Introduction

Data visualizations — also known as graphs, charts, and plots — encode quantitative information using a combination of graphical elements, numerals, and words. They are powerful ways of helping people to rapidly grasp quantitative patterns and trends across many fields, including science, journalism, and business (Franconeri et al., 2021). To build AI systems that can operate effectively in all of these contexts will require them to be able to parse a knowledge base that blends graphics and text. However, current vision-language models (VLMs) have not yet reached human-level performance on even basic data visualization understanding tasks (Verma et al., 2024; Pandey & Ottley, 2025). This performance gap raises the question motivating the current work: when and why do VLMs fail to parse data visualizations?

Many current VLMs consist of multiple modules that each operate over text and image inputs separately, though different VLMs embody different architectural commitments as to how to combine latent information from visual and linguistic inputs (Grattafiori et al., 2024; Deitke et al., 2024; Team, 2024; Chen et al., 2024; Laurençon et al., 2024; Lu et al., 2024). For any given VLM, the failures might thus stem from multiple potential sources: the visual component (e.g., failure to encode the graph), the language component (e.g., failure to reason over the visual information), or the transfer of

---

[1]Dataset & code are available at https://github.com/cogtoolslab/fugu.

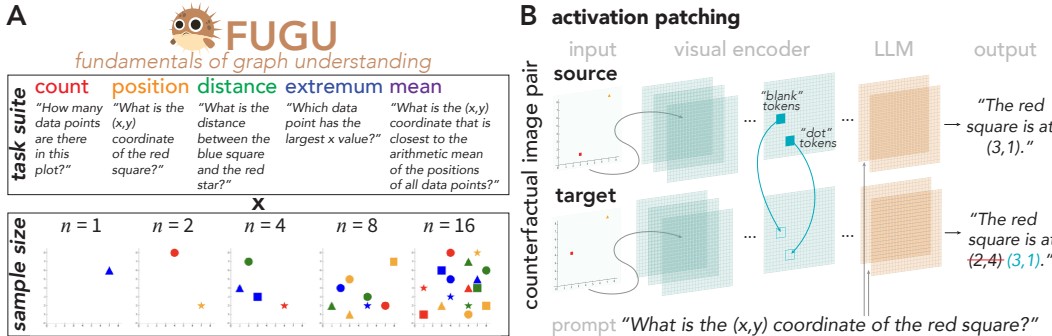

Figure 1: **A) The Fundamentals of Graph Understanding (FUGU) task suite and dataset.** The top row (task suite) displays example prompts for each of our five tasks. The bottom row (sample size) shows example scatter plots from our dataset for each number of data points ($n$). FUGU pairs the five tasks with all applicable scatter plots, resulting in 3,968 unique <task, image> pairs. **B) Visualization of the causal intervention method.** We perform causal interventions on a counterfactual pair of scatter plots by passing each separately through the model with the same prompt, extracting embeddings from the vision encoder, and swapping the visual embeddings corresponding to a meaningful subset of image tokens (like a data point) from the source to the target run. If this successfully replaces model output behavior on target with the source prediction, then we have located a causally important part of the representation for that task.

information between the vision and language components (e.g., failure to align visual and linguistic latent spaces). However, distinguishing between these different failure modes is not trivial given the complexity of many multimodal reasoning tasks and many prevalent model architectures.

To more precisely identify these potential points of failure in data visualization understanding, we developed FUGU (*Fundamentals of Graph Understanding*). FUGU is a novel benchmark consisting of 3,968 questions paired with scatter plots designed to probe the foundational spatial reasoning and mathematical skills that are needed to parse any data visualization, including the ability to report the values of data points, estimate distances between them, and estimate summary statistics, including the minimum, maximum, and mean. FUGU consists of entirely novel, synthetically generated visualizations that enable more targeted investigation of these foundational skills, allowing us to diagnose where a model's understanding breaks down.

Towards this end, we leveraged multiple mechanistic interpretability techniques to locate the information bottlenecks that hinder model performance. We use activation patching (also known as causal interventions; Vig et al. 2020; Finlayson et al. 2021; Geiger et al. 2021; 2024; Meng et al. 2022; Wang et al. 2023) to test whether a specific component of a model encodes behaviorally relevant information. In addition, we use linear probes — lightweight classifiers on internal model representations — to evaluate the presence of task-relevant information (e.g., the location of a data point).

While this approach could be applied to any VLM architecture, in this work we conducted a thorough investigation of three models: LLaMA-3.2B 11B (Grattafiori et al., 2024), LLaVA-OneVision 7B (Li et al., 2024), and InternVL3 14B (Zhu et al., 2025). These VLMs are widely used, performant, and vary along key design choices: architecture type (cross-attention based vs. decoder-only), vision encoder, LLM backbone, and pretraining data. Our findings serve to identify the limitations of existing VLMs and provide insight towards building better VLMs.

## 2 RELATED WORK

A growing body of work at the intersection of computer vision and visualization aims to develop AI systems that understand data visualizations as well as humans can. Our paper builds most directly on prior work benchmarking data visualization understanding in AI systems, as well as recent advances in mechanistic interpretability methods applied to large language models and vision-language models.

**Machine data visualization understanding**  Several visual question answering benchmarks currently exist to assess progress towards this goal, including FigureQA (Kahou et al., 2017), DVQA (Kafle et al., 2018), PlotQA (Methani et al., 2020), ChartQA (Masry et al., 2022), and ChartQA-Pro (Masry et al., 2025). While early efforts like FigureQA found initial success on synthetic data visualization understanding tasks, evaluations were conducted exclusively with models only capable of drawing from a small, fixed vocabulary (i.e., true/false or color names). Subsequently developed benchmarks, such as DVQA and PlotQA, introduced a wider variety of more complex plots and questions that required reasoning over real-valued quantities and generating responses from an unbounded vocabulary. Many items in ChartQA require arithmetic operations to be performed over the plotted values, such as addition or subtraction, which have posed persistent challenges for many otherwise performant VLMs (Golovanevsky et al., 2024). More recent work has found that these limitations extend to benchmarks initially developed to assess human data visualization understanding, suggesting they are not specific to ChartQA (Verma et al., 2024).

**Interpretability of vision-language models**  The current paper applies recently developed methods to localize the mechanisms that are responsible for VLM behavior (Allen et al., 2025; Liu et al., 2025; Ho et al., 2025). In particular, this work makes use of a combination of activation patching (Vig et al., 2020; Finlayson et al., 2021; Geiger et al., 2021; 2024; Meng et al., 2022; Wang et al., 2023) and linear probing methods (Golovanevsky et al., 2024; Dahlgren Lindström et al., 2020; Hendricks & Nematzadeh, 2021; Cao et al., 2020) to locate the internal components within neural network models that are responsible for the patterns of success and failure displayed by VLMs.

## 3 METHODS

### 3.1 DATASET

We developed FUGU, a task suite and dataset containing 3,968 questions paired with 768 scatter plots (Figure 1). Scatter plots are an important case study because they jointly rely on fundamental visual operations (*"to know what is where by looking"*; Marr 2010) and processing of symbolic inputs, such as labeled axes and numerals. In addition, they afford opportunities to see how those competencies interact with foundational mathematical reasoning skills, including counting and basic arithmetic.

**Plot generation**  The plots in FUGU were procedurally generated to ensure precise experimental control over the complexity and appearance of each plot. The scatter plots contain $n \in \{1, 2, 4, 8, 16\}$ data points. To ensure that a concise natural-language expression could always be used to unambiguously refer to each data point, each observation was represented by a unique shape-color combination, drawn from a fixed set of four shapes (circle, triangle, square, star) and four colors (red, blue, green, yellow). The plots were rendered with a white background and black axes, with integer tick marks and labels ranging from 0 to 8 on both axes. To avoid occlusion of one data point by another, we constrained all data points to occupy non-overlapping natural-number positions in the 8×8 coordinate plane, and the positions of individual data points were randomly sampled to generate each plot. For plots with $n = 1$, we sampled 4 shape-color configurations at each of the 64 possible positions, yielding 256 unique images. For $n = 2$, we fixed one object at position $(1, 1)$ and exhaustively positioned a second object at each of the remaining 63 positions, randomizing the shapes and colors for each coordinate combination. This ensures that all possible distances between two data points are represented. We then positioned pairs of objects at 65 randomly selected, non-overlapping positions to generate a total of 128 $n = 2$ plots. For each of $n \in \{4, 8, 16\}$, we randomly sampled 128 unique position configurations. These design choices make FUGU visualizations quite simple, but FUGU turns out to be extremely difficult for present-day VLMs (as we show in Section 4).

**Task suite**  Consistent with previous work on data visualization literacy (Lee et al., 2016; Börner et al., 2019; Boy et al., 2014), we administered the following five basic tasks:

1. **Count**: Identifying the total number of data points present in the visualization.

2. **Position**: Reporting the exact coordinates of a specific data point.

3. **Distance**: Calculating the Euclidean distance between two data points.

4. **Extremum**: Identifying data points with minimum or maximum values along the $x$ or $y$-axis.

5. **Mean**: Computing the arithmetic mean of all data point positions along the $x$ and $y$-axis.

We administered the count and position tasks paired with all 768 scatter plots, but the distance task was only used with plots containing $n \geq 2$, and the extremum and mean tasks with plots containing $n \geq 4$. This yielded 3,968 unique <task, image> pairs distributed across the five task categories.

## 3.2 MODELS

In this work, we investigated LLaMA-3.2 11B, LLaVA-OneVision 7B, and InternVL3 14B.[2] Each model consists of three major components: the vision encoder, the language model (LM), and the vision-language adapter. Visual representations are first extracted from a given model's vision encoder, projected into the same dimensionality as the LM via the adapter, and then passed into the LM either as either the keys and values in cross-attention layers (Alayrac et al., 2022) throughout the model (LLaMA-3.2) or as additional tokens in the input prompt (LLaVA-OneVision and InternVL3).

The models vary along a number of key dimensions. Each model employs a unique vision encoder: a CLIP-pretrained vision transformer (ViT; Radford et al. 2021; Dosovitskiy et al. 2020) for LLaMA-3.2, a SigLIP (Zhai et al., 2023) ViT for LLaVA-OneVision, and a custom InternViT for InternVL. While the first two vision encoders are standard ViTs, InternViT employs a multicrop strategy whereby four higher-resolution crops of the original input image are also passed through the encoder. The models also use different LM backbones: while LLaVA-OneVision and InternVL both use Qwen (Qwen2-7b and Qwen2.5-7b, respectively; Yang et al. 2024; Bai et al. 2025), LLaMA-3.2 uses LLaMA 3.1 (Dubey et al., 2024).

## 3.3 MODEL EVALUATION PROTOCOL

Text input to a given model consists of two components: general context, followed by a specific question (see Figure 1 for examples). The context identifies the visual input as a scatter plot with $x$ and $y$-axes and indicates the four possible shapes and colors that data points can have (see Appendix A.1 for full text). Models were explicitly instructed to round their answers to the nearest integer when reporting coordinates, but not given any additional guidance or constraints. All model responses were generated with temperature set to 0 and a maximum output length of 1,000 tokens.

For evaluation, we employed Claude-3.5 Sonnet as an automated judge. The judge received the complete model response along with the ground-truth answers and was tasked with extracting the model's predicted answers and determining if they match the ground-truth values (see Appendix A.2 for the full evaluation prompt).[3] Our evaluation criteria accommodate reasonable variations in response format: for distance and mean tasks, we accepted values rounded either up or down to the nearest integer. However, counting and position tasks required exact matches to be considered correct. For extremum tasks (identifying minimum or maximum points), we implemented flexible matching criteria. A response was considered correct if it identified both the shape and color of the target object, or if it uniquely identified the object by specifying only one attribute that is sufficient for disambiguation. For example, if the minimum point along the y-axis was a green square and it was the only green object in the plot, the answer "green" was considered correct.

## 3.4 CAUSAL INTERVENTIONS

To isolate the representations in the visual encoder responsible for producing any observed pattern of model success and failure, we include a number of causal interventions on the visual encoder representations corresponding to the spatial locations of the dots in the image. To perform an intervention, we took a set of *source* activation vectors from the ViT layer outputs, denoted individually as $h^{src}_{(r,c,\ell)}$, from row $r$, column $c$, and layer $\ell$ of the representations, and we use them to replace the corresponding *target* activations $h^{trg}_{(r,c,\ell)}$ at the same row, column, and layer of the ViT under a different input image. The model then uses this intervened target representation to continue its processing. For a given layer, we extracted source activations corresponding to minimally sufficient

---

[2]For convenience, we drop the parameter count when referring to these models throughout the paper.

[3]We verified Claude 3.5 Sonnet's reliability as a judge by comparing its judgments to a hand-designed regex. On items the regex could parse, we found 99% agreement with Claude 3.5 Sonnet extractions.

rows and columns that contained the dot(s) in the source image and the minimal rows and columns that contain the dot(s) in the target image.[4]

### 3.5 LINEAR PROBES

To determine whether task failures stem from visual encoding deficiencies or issues with the vision-language connection, we employed linear probes to assess how information is represented across model components.[5] We trained linear probes at each layer of the vision encoder and the LLM separately for task-relevant information, such as the coordinates of individual data points or the distances between data points (see Appendix F.1 for details). This approach helps distinguish between "perceptual" failures where the vision encoder failed to represent the information from the visualization from "extraction" failures where the information may have been present in the visual representation but the LLM failed to parse it.

For the position task, we trained 16 distinct probes per layer, each predicting the $(x, y)$ coordinates of one of 16 specific objects (e.g., "red square") (see Appendix F.2). The target data points in the training dataset covered all $x$ and $y$-coordinates but not all $(x, y)$ combinations — each $x$-coordinate is paired with 6 possible $y$-coordinates, while holding out the remaining 16 $(x, y)$ combinations for testing. Probes were trained for 5,000 epochs with a constant learning rate of $1 \times 10^{-3}$ on 1,200 training images and evaluated on 400 test images.

## 4 RESULTS

### 4.1 MODEL PERFORMANCE WORSENS AS THE SAMPLE SIZE INCREASES IN THE PLOT

Although FUGU tasks are simple, they present a nontrivial challenge for models: InternVL3 demonstrates the highest accuracy averaged across the items (69.3%), followed by LLaVA-OneVision (55.7%) and LLaMA-3.2 (55%). Furthermore, we found that model performance deteriorates as the number of data points increases (Figure 2). Accuracy on the counting task exhibits the most dramatic decline, falling from 100% with a single point to 0% with 16 points for all three models. Similarly, position identification accuracy drops considerably, although InternVL3 maintains rather high performance on this task for a number of points less than 16 (>80%). These results confirm that FUGU is a useful diagnostic tool for identifying model bottlenecks on fundamental data visualization understanding tasks. See Appendix B for benchmarking results on a larger set of VLMs and Appendix E for error patterns.

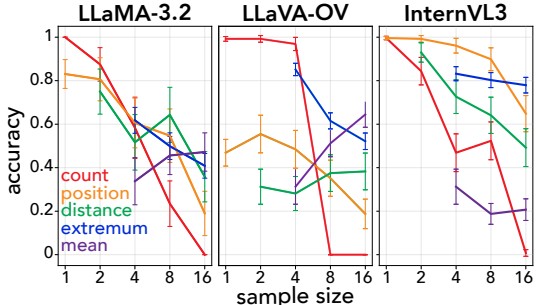

Figure 2: **Model behavioral performance on FUGU.** The $x$-axis represents the number of data points in each scatter plot, and the $y$-axis shows the accuracy of freely-generated VLM responses. Error bars show 95% CIs.

### 4.2 TASK-RELEVANT INFORMATION BECOMES MORE DISTRIBUTED ACROSS TOKENS IN DEEPER LAYERS OF VISION ENCODER

To understand which image features drive successful performance across our task suite, we conducted a series of causal interventions on each model's vision encoder (Figure 1B). By running a given model on minimally different pairs of images and swapping out the visual representations on

---

[4]Before the first ViT attention block, these interventions are equivalent to trivially replacing the target image with the source image.

[5]Linear probes provide an optimistic estimate of information availability: if a probe fails to extract certain information, we can conclude that the information is not *linearly* decodable from the representations. However, probe success does not indicate how the model uses this information, as the probes may leverage features that the model ignores or uses for unrelated computations.

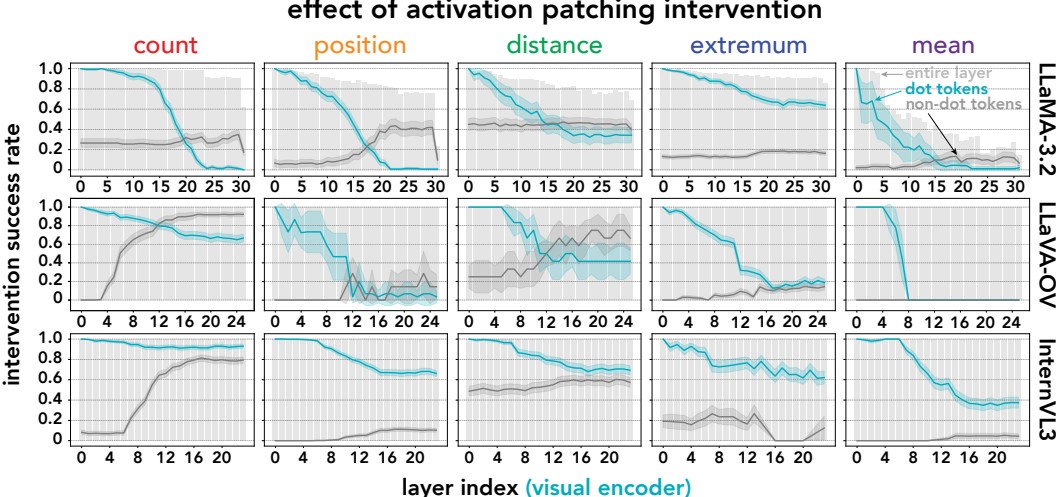

Figure 3: **Success rates for causal interventions on the vision encoders of LLaMA-3.2** (top row)**, LLaVA-OV** (middle row)**, and InternVL** (bottom row)**.** Each individual plot shows success rates for causal interventions on a particular subset of image tokens at a particular vision encoder layer for each of the five tasks (indicated by the labels above the columns). The $x$-axis corresponds to vision encoder layers, while the $y$-axis shows the proportion of counterfactual pairs where the output answer on the target input was successfully swapped to the expected source output as a result of the intervention. The gray bars labeled "full layer" show baseline intervention accuracy for swapping the entire layer. The line labeled "dot" shows success rates when only intervening on tokens containing data points, while the line labeled "other" shows success rates for intervening on all non-dot tokens (including the CLS token).

meaningful subsets of image tokens (such as all tokens containing data points), we sought to isolate the representational components that are responsible for the model's response to each question.

Our experiments revealed that in the earliest visual layer (layer 0), interventions on data point ("dot") tokens yielded a 100% intervention success rate across all tasks and models, suggesting that all task-relevant information is initially concentrated in the representations registered to the positions of the data points in the image (Figure 3). Across vision encoder layers, we found a general trend: task-relevant information tends to become increasingly distributed across tokens within each layer. Specifically, the causal importance of just the "dot" tokens in each of the deeper layers decreases relative to that of all tokens in that layer, while the importance of the "non-dot" tokens in deeper layers remains relatively stable or increases. Notably, the performance achieved by full-layer interventions (gray bars in Figure 3) consistently exceeds what could be attained by independently intervening on either data point or non-data point tokens alone. This indicates that successful task completion depends on the interaction of information between different visual components.

Overall, these findings are consistent with the notion that models employ distributed representations to contextualize information initially contained in the most information-dense regions of the input to support a broad variety of downstream tasks.

### 4.3 ERRORS IN ESTIMATING $(x, y)$ COORDINATES ARE ESPECIALLY COMMON FOR MORE COMPLEX PLOTS

To better understand the pattern of results in Section 4.1, we analyzed each model's problem-solving approach by analyzing the chain-of-thought reasoning traces it generated. We discovered that LLaMA-3.2 and InternVL3 frequently employed a strategy of explicitly listing the $(x, y)$ coordinates of data points in their chain-of-thought before performing arithmetic operations on these values (see

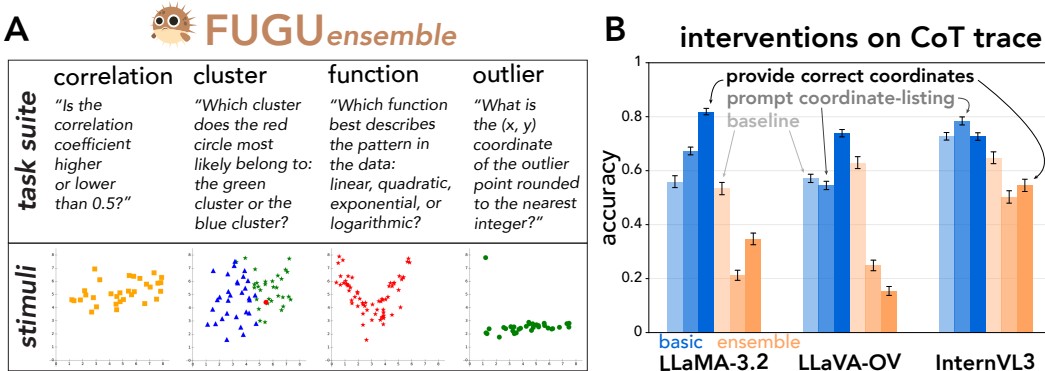

Figure 4: **A) Examples of ensemble task stimuli and prompts. B) Providing ground truth points in the chain-of-thought improves performance for most tasks.** The palest bars for each task show baseline behavioral accuracy when the model is allowed to freely generate (i.e. results from Figure 2). The blue "basic" bars show accuracy averaged across the basic FUGU tasks (see Figure 1), while the orange "ensemble" bars show accuracy averaged across ensemble tasks (see Section 4.6). The middle bars show accuracy when the model is provided with its own generated coordinates for each data point in the scatter plot as part of its chain-of-thought, while the darkest bars show accuracy when models are given ground-truth coordinates. The performance gains offered by the ground-truth listing suggests that accurate coordinate extraction is a significant bottleneck.

Appendix C.1 and C.5).[6] Any inaccuracies in extracting these coordinates might act as a bottleneck, propagating errors throughout the reasoning chain and ultimately leading to an incorrect response.

To investigate this hypothesis, we evaluated the accuracy with which models could extract coordinates. For each scatter plot, we prompted a given model to list the coordinates of every individual data point. We found that while for the simplest plots, LLaMA-3.2 achieved an accuracy of 91%, but coordinate generation declined to 20% accurate for more complex plots containing more data points (Figure 6 in Appendix C.2). LLaVA-OneVision's performance peaks for plots of two data points (60% accuracy), similarly degrading to 20% as the number of data points increases. On the other hand, InternVL's coordinate generation accuracy remains near ceiling even for plots with many data points. These findings suggest that errors in coordinate extraction might be associated with model difficulty with more complex displays (Figure 2).

### 4.4 PROVIDING CORRECT $(x, y)$ COORDINATES IMPROVES PERFORMANCE

To test the hypothesis that accurate coordinate listing is an important source of difficulty for models, we conducted two interventions on their chain-of-thought traces. The first approach was to explicitly prompt the model to list the coordinates for all data points as the first step in reasoning before answering the question. The second was to provide the correct coordinates to the model as the starting point for reasoning (see Appendix C.3 for details).

We found that providing ground-truth coordinates substantially improved performance across tasks for LLaMA-3.2 and LLaVA-OneVision ( Figure 4B). In contrast, reasoning from the model's own extracted coordinates either yielded comparatively small performance gains or degraded performance. InternVL performance remains relatively stable across interventions and is highest for the model-listing condition, likely owing to its highly accurate coordinate extraction (Section 4.3). Overall, these results confirm that accurate coordinate extraction is indeed a key limitation for VLMs.

Notably, even with ground-truth coordinates provided, perfect accuracy remained elusive across tasks. This finding suggests other limitations in model abilities to effectively use this coordinate information,

---

[6]LLaVA-OneVision tends to directly output answers instead of performing chain-of-thought. However, providing LLaVA with a list of ground-truth $(x, y)$ coordinates in its context substantially improves performance, suggesting that accurately extracting points is also a bottleneck for this model. See Section 4.4.

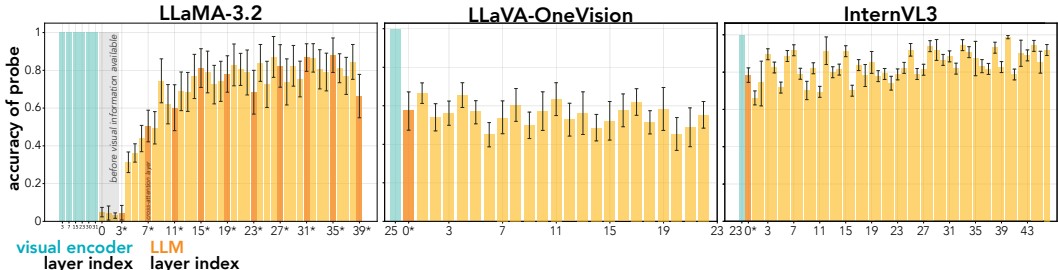

Figure 5: **Probe test accuracy for position in vision vs. language layers.** In each panel, the $x$-axis corresponds to model layer for the vision encoder (left, blue) and LLM (right, orange), while the $y$-axis shows mean probe test accuracy for position on that layer. Only vision layers that connect to the LLM (either via cross-attention in LLaMA-3.2 or as a direct input to the LLM context in LLaVA-OneVision and InternVL3) are shown. The asterisked LLM layers correspond to language blocks that receive visual inputs (again, either via cross-attention in LLaMA-3.2 or as input tokens for the other models).

possibly due to challenges in maintaining the association between the listed coordinates and their corresponding data points or in executing subsequent reasoning steps.

### 4.5 INFORMATION ABOUT CORRECT POSITIONS IS REPRESENTED THROUGHOUT THE VISUAL ENCODER

Is the failure to list the correct $(x, y)$ coordinates as reported in Sections 4.3 and 4.4 due to a deficiency of the vision encoder, or a problem in the LLM or handoff to the LLM? To disentangle these possibilities, we trained linear probes on the visual encoder layers and LLM layers to predict $(x, y)$ coordinates for each object (see Section 3.5). If the probes fail to predict coordinates from both vision and language representations, this would indicate that the information is simply not represented in the vision encoder, and therefore cannot be accessed by the LLM. On the other hand, if the information can be decoded from the vision encoder but not the LLM, this suggests that the connection to the LLM is the bottleneck.

We found that position is decodable with 100% test accuracy across all data points in the vision encoder for each model (Figure 5). Probe test accuracy increases gradually throughout LLaMA-3.2's LLM layers — exhibiting a noticeable jump in accuracy after the model's first cross-attention layer — but is never at the same level it is in the visual encoder. On the other hand, LLaVA-OneVision exhibits a sharp drop in probe test accuracy within its LLM layers that does not recover, while InternVL3 maintains relatively high accuracy throughout.

The gap in probe test accuracy between vision and language layers across all three models suggests that the LLM performs additional, non-linear processing on the visual encoding to extract coordinates, despite the coordinates being linearly accessible in the vision encoder. These findings suggest a potential inefficiency in the hand-off from the vision to the language component of these models which may impact performance.

### 4.6 POINT-LISTING DOES NOT GENERALIZE TO MORE COMPLEX TASKS

Our findings in Sections 4.3 and 4.4 revealed that LLaMA-3.2 and InternVL3 frequently employ a coordinate listing strategy when solving FUGU tasks, and that for all three models, providing ground-truth coordinates substantially improves performance across most tasks. If simply providing lists of extracted $(x, y)$ coordinates to the LLM also improves performance on more naturalistic plots and realistic tasks that require reasoning over many more data points, then this may present a viable solution for data visualization understanding with VLMs.

To investigate the generalizability of our findings to more complex tasks, we developed an extended dataset featuring scatter plots with higher data point densities (16, 32, 64, and 128 points) paired

with four tasks that require ensemble-level reasoning rather than individual point identification (see Figure 4A):

1. **Correlation**: Judging whether the correlation coefficient between $x$ and $y$ is above or below 0.5.

2. **Cluster**: Identifying which of two clusters a query point is most likely to belong to.

3. **Function**: Determining which function family (linear, quadratic, exponential, or logarithmic) best fits the data.

4. **Outlier**: Identifying the coordinate of the data point that represents an outlier in the distribution.

We replicated our experimental approach from Section 4.4 on these "ensemble tasks," testing the same three conditions: free generation, prompted coordinate listing, and provision of ground-truth coordinates in the model's context. In the free generation condition, models never spontaneously adopted the coordinate listing strategy that proved central to their approach on basic FUGU tasks. Furthermore, providing ground-truth coordinates consistently harmed performance across all ensemble tasks and point densities. This performance degradation suggests that while coordinate listing serves as an effective strategy for basic spatial reasoning tasks, it fails to generalize to more complex tasks that require spatial reasoning over many points — thus, it does not represent a general solution for VLM data visualization understanding.

### 4.7 FINE-TUNING IS INSUFFICIENT TO OVERCOME THE CHALLENGES POSED BY FUGU

Is poor model performance on the basic FUGU tasks and the ensemble tasks from Section 4.6 simply a consequence of insufficient exposure to data visualization tasks during pretraining? If so, targeted fine-tuning on visualization data should substantially ameliorate these performance gaps.

| Model ↓ | Base | Fine-tuned |
|---|---|---|
| LLaMA-3.2 | 54.2 | 77.7 (+23.5) |
| LLaVA-OV | 59.4 | 77.9 (+18.5) |
| InternVL3 | 67.1 | 85.9 (+18.8) |

Table 1: **Test accuracy (%) on FUGU ($100k$) for base models vs. fine-tuned models.** Fine-tuning improves performance by up to 23.5 points, but none of the fine-tuned models reach ceiling test accuracy. Accuracy broken down by task can be found in Appendix B.

To test this possibility, we fine-tuned all three models on training datasets that combine the five FUGU tasks (count, position, distance, extremum, mean) with the four ensemble tasks (correlation, cluster, function, outlier). We varied the training dataset size $\in \{10k, 100k\}$ samples. We also conducted a large hyperparameter sweep over relevant dimensions (learning rate, scheduler, and optimizer). Training details can be found in Appendix D.

Evaluation of the finetuned models on the original FUGU and ensemble test revealed that, while performance improved substantially compared to the pretrained baselines, none of the models achieved ceiling performance on either task suite (Table 1), even at the largest training data size. These findings suggest that the visualization understanding limitations observed in our experiments are unlikely to be due solely to insufficient training data exposure.

## 5 DISCUSSION

In this work, we investigated bottlenecks in data visualization understanding by vision-language models (VLMs). Towards this end, we created FUGU, a task suite and dataset that poses fundamental challenges for reasoning about multimodal inputs. Using behavioral evaluation, causal interventions, and linear probes, we discovered that a primary bottleneck for these VLMs lies not in their visual encoding or mathematical reasoning capabilities, but in the hand-off between the vision and language components. While the vision encoder successfully represents data point coordinates and spatial information (as demonstrated by high linear probe accuracy), the language model component often struggles to access and use this information effectively. On the other hand, the language model component seemed to be more successful in extracting task-relevant visual information when it was represented in a more localized manner within the visual encoder (as it was for InternVL3).

Our results also point toward several promising directions for improving VLMs. First, current multimodal connectors are typically pretrained on image-caption objectives, which offer little incentive to preserve fine-grained spatial structure. Connectors that are explicitly optimized to extract and

maintain relational and geometric information may mitigate degradation of this information during the vision-language handoff. Second, given the richness of visual representations (as confirmed by our probes), it is unlikely that the same extraction strategy is optimal for every task. While basic FUGU tasks may require very fine-grained spatial detail, higher-level ensemble tasks may instead benefit from representations that foreground global structure or higher-level shape information. This suggests the value of dynamic, prompt-conditioned connectors that adapt how they attend to and transform visual features depending on the task described in the query, rather than relying on a single, static projection for all downstream reasoning. Such adaptive interfaces could help ensure that the most optimal level of task-relevant spatial features are accessible to the language model. Finally, our findings may motivate architectures in which reasoning does not occur exclusively in the language module. Recent advances in generative vision models indicate that models can increasingly perform structured problem solving by generating intermediate visual states (i.e. frames) rather than text tokens (Wiedemer et al., 2025; Xu et al., 2025). Allowing VLMs to reason within the latent space of the vision module—e.g., by generating intermediate visual states—may help bypass lossy transformations at the vision-language boundary entirely and support more robust multimodal reasoning on tasks that rely critically on spatial structure.

Taken together, our findings highlight the value of seemingly simple multimodal reasoning tasks (i.e., answering questions about scatter plots) for identifying failure modes in current VLMs. Moreover, they showcase the critical role that mechanistic interpretability techniques can play in identifying the *sources* of failure within these models, where behavioral evidence alone would have been insufficient. Whereas previous applications of these techniques to multimodal models have been performed on much simpler tasks (e.g., same vs. different judgments) and models (e.g., CLIP-trained encoders; Lepori et al. (2024)), here we use a similarly experimentally rigorous approach to study a more diverse suite of naturalistic multimodal reasoning tasks and the internal mechanisms of generative models with more complex and varied architectures. In sum, this work suggest key opportunities for further investigation of the mechanisms that enable effective multimodal reasoning in AI systems.

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

## A  FULL PROMPTING DETAILS

### A.1  BEHAVIORAL EVALUATION PROMPT

All behavioral evaluation input prompts begin with a standardized paragraph describing the input image in a generic way. This is meant to provide additional helpful context for the model without explicitly guiding the chain-of-thought. The text is the following:

> The image shows a scatter plot displaying the relationship between two quantitative variables, labeled 'x' (horizontal axis) and 'y' (vertical axis). Each observation in the dataset is represented by a single graphical element (circle, triangle, square, or star) positioned on the coordinate plane according to its exact x- and y-values. The data points appear in four distinct colors (red, green, blue, or orange). Please answer the following question based on the information conveyed by this scatter plot.

### A.2  INPUT PROMPT TO THE LLM JUDGE

As described in Section 3.3, we employ Claude 3.5 Sonnet to evaluate the model responses on FUGU (results reported in Section 4.1 and Figure 2). The full prompt submitted to Claude for each <task, image> pair in FUGU is the following:

> I have asked a vision-language model to answer a {task_type} question about a plot. Here is the question:
>
> "{question}"
>
> The ground truth answer (or possible answers, formatted as a list) to the question are: {answer}. The model is correct if it returns an answer that is in the set of possible answers.
>
> This is the full response from the model:
>
> "{full_response}"
>
> Based on the type of task, I expect that the answer {answer_spec}. The model may have responded with an answer that is not fully formatted correctly. For example, it may answer "the star symbol" instead of specifying a color. It may also omit the first parenthesis in a parenthesis pair. Finally, the model may have forgotten to round the numbers to the nearest integer. If this answer matches one of the possible answers, it is still correct.
>
> Please analyze the model's response and extract an answer that matches the expected format above. Try to be as faithful as possible to the model's response while still matching the expected format. Numbers may be provided as words in the response, and you should convert them to numbers. If an answer is able to be extracted, please also analyze if it is correct by comparing it to the ground truth answer(s). If no good answer can be found, please explain why.
>
> Return your analysis in this format:
>
> Extracted Answer: [number or 'None']
>
> Correct: [True/False]
>
> Explanation (if 'None'): [your reasoning]

The templated values in pink are filled according to the specific <task, image> response that is being evaluated.

- {task_type}: count, position, distance, extremum, mean
- {question}: the task prompt (see the top row of Figure 1 for examples)
- {answer}: a comma separated list of all possible ground-truth answers to the task
- {full_response}: the entire decoded text of LLaMA-3.2 11B's output (maximum 1,000 generated tokens)
- {answer_spec}: a description of the expected output type. For count and distance tasks, this is "is a single integer." For position and mean: "is an $(x, y)$ coordinate enclosed by parentheses." For extremum tasks: "is a (color, shape) combination."

Claude was able to successfully extract properly-formatted answers for LLaMA-3.2 11B on 100% of the inputs, so our behavioral results include responses on the full dataset.

## B  BENCHMARKING ADDITIONAL VLMs ON FUGU AND ENSEMBLE TASKS

We additionally benchmarked the following VLMs on FUGU and the ensemble tasks from Section 4.6: Molmo 7B (Deitke et al., 2025), Gemma 3 12B (Team et al., 2025), Idefics2 (Laurençon et al., 2024), Chameleon 7B (Team, 2024), Claude 3.5 Sonnet and Haiku, GPT-4o mini, GPT-4o, GPT-5, Gemini 2.5 Flash. Results for FUGU tasks are in Table 2; results for ensemble tasks are in Table 3.

| Model ↓ | Count | Position | Distance | Extremum | Mean | Avg. |
|---|---|---|---|---|---|---|
| LLaMA-3.2 11B | 63.5 | 63.9 | 56.6 | 50.8 | 43.5 | 55.7 |
| LLaMA-3.2 11B (FUGU-100$k$) | 99.3 | 91.7 | 87.3 | 68.1 | 50.5 | 79.4 |
| LLaVA-OneVision 7B | 65.8 | 41.9 | 33.8 | 66.3 | 52.5 | 52.1 |
| LLaVA-OneVision 7B (FUGU-100$k$) | 86.8 | 88.0 | 36.3 | 65.2 | 94.2 | 74.1 |
| InternVL3 14B | 63.9 | 91.5 | 69.7 | 80.5 | 22.0 | 65.5 |
| InternVL3 14B (FUGU-100$k$) | 100.0 | 99.2 | 87.7 | 88.5 | 93.3 | 93.7 |
| Molmo-D 7B | 39.8 | 54.6 | 37.9 | 72.9 | 32.8 | 47.6 |
| Molmo-O 7B | 69.1 | 45.3 | 34.2 | 52.2 | 30.8 | 46.3 |
| Gemma3 12B | 70.1 | 90.5 | 84.8 | 87.7 | 86.1 | 83.8 |
| Idefics2 8B | 49.0 | 3.4 | 34.0 | 31.4 | 9.7 | 25.5 |
| Chameleon 7B | 0.0 | 2.2 | 20.1 | 13.7 | 2.7 | 7.7 |
| Claude 3.5 Sonnet | 86.6 | 95.3 | 97.1 | 97.5 | 75.3 | 90.4 |
| Claude 3.5 Haiku | 86.2 | 95.7 | 97.1 | 97.7 | 74.5 | 90.2 |
| GPT-4o mini | 69.4 | 63.9 | 61.3 | 68.9 | 65.3 | 65.8 |
| GPT-4o | 86.3 | 86.3 | 81.8 | 93.8 | 83.0 | 86.2 |
| GPT-5 | 99.2 | 100.0 | 99.8 | 69.6 | 19.7 | 77.7 |
| Gemini 2.5 Flash | 99.5 | 99.6 | 100.0 | 96.4 | 81.6 | 95.4 |

Table 2: **Model behavioral performance on FUGU tasks**. See Figure 1A for examples of each task. For each task, model accuracy is averaged across sample size $\in \{1, 2, 4, 8, 16\}$. The "**Avg.**" column gives accuracy averaged across all five tasks.

| Model ↓ | Correlation | Cluster | Function | Outlier | Avg. |
|---|---|---|---|---|---|
| LLaMA-3.2 11B | 74.6 | 74.8 | 26.5 | 34.5 | 52.6 |
| LLaMA-3.2 11B (FUGU-100$k$) | 82.3 | 90.4 | 52.1 | 86.8 | 77.9 |
| LLaVA-OneVision 7B | 69.0 | 82.9 | 49.2 | 48.5 | 62.4 |
| LLaVA-OneVision 7B (FUGU-100$k$) | 76.7 | 93.8 | 69.8 | 87.0 | 81.8 |
| InternVL3 14B | 55.8 | 91.5 | 44.8 | 67.5 | 64.9 |
| InternVL3 14B (FUGU-100$k$) | 71.0 | 92.5 | 65.2 | 87.2 | 79.0 |
| Molmo-D 7B | 51.5 | 69.8 | 37.3 | 42.5 | 50.3 |
| Molmo-O 7B | 50.2 | 47.9 | 25.0 | 0.0 | 30.8 |
| Gemma3 12B | 57.3 | 85.6 | 48.5 | 77.5 | 67.2 |
| Idefics2 8B | 66.7 | 55.6 | 40.0 | 4.2 | 41.6 |
| Chameleon 7B | 50.6 | 54.8 | 24.6 | 0.0 | 32.5 |
| Claude 3.5 Sonnet | 81.7 | 89.6 | 71.0 | 66.5 | 77.2 |
| Claude 3.5 Haiku | 84.0 | 90.0 | 69.6 | 69.8 | 78.4 |
| GPT-4o Mini | 82.3 | 88.8 | 57.9 | 20.0 | 62.2 |
| GPT-4o | 84.8 | 94.8 | 76.5 | 56.8 | 78.2 |
| GPT-5 | 82.9 | 93.5 | 80.8 | 81.5 | 84.7 |
| Gemini 2.5 Flash | 81.2 | 94.2 | 71.0 | 89.0 | 83.8 |

Table 3: **Model behavioral performance on ensemble tasks from Section 4.6**. See Figure 4A for examples of each task. The "**Avg.**" column gives accuracy averaged across all four tasks.

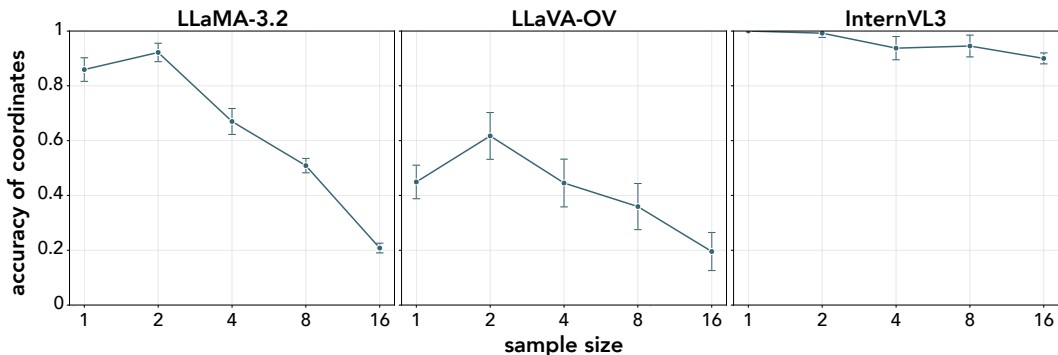

Figure 6: **Accuracy of model-generated $(x, y)$ coordinates**. The $x$-axis represents the number of data points in each scatter plot, while the $y$-axis shows the accuracy of model-generated coordinates. We prompted models to generate coordinates for each element in the plot.

## C EXAMINING MODEL CHAIN-OF-THOUGHT ON FUGU TASKS

### C.1 FREQUENCY OF POINT LISTING BEHAVIOR

How often do models list individual data point coordinates in order to solve tasks? We analyzed the frequency of point listing behaviors by passing full model responses from Section 4.1 to Claude 3.5 Sonnet, asking Claude to judge whether the response contained any lists of coordinates ($x$, $y$, or $(x, y)$).

The resulting coordinate-listing frequencies are shown in Table 4. For LLaMA-3.2 and LLaVA-OneVision, the frequency of coordinate listing is extremely variable across tasks. None of the count tasks trigger point listing for either model, while almost all of the mean tasks do. Meanwhile, InternVL lists coordinates most of the time for most tasks. These findings provide context for the performance patterns observed in Figure 4B, reinforcing our hypothesis that accurate coordinate extraction represents a fundamental bottleneck in data visualization understanding for these models.

| Task: | count | position | distance | extremum | mean |
|---|---|---|---|---|---|
| LLaMA-3.2 | 0% | 98% | 100% | 53% | 100% |
| LLaVA-OneVision | 0% | 100% | 0% | 17% | 100% |
| InternVL3 | 70% | 100% | 100% | 99% | 84% |

Table 4: **Frequency of point listing (%) in responses generated by models by task.** Each cell contains the percentage of model responses that contain point listing as judged by Claude 3.5 Sonnet.

### C.2 ACCURACY OF POINT LISTING

All three models demonstrate a degradation in performance on coordinate extraction as the number of data points increases. See Figure 6.

### C.3 STRUCTURED CHAIN-OF-THOUGHT: MODEL VERSUS GROUND-TRUTH COORDINATE LISTING

We forced models to implement one of two strategies by providing a given model with structured chain-of-thought templates. The input prompts follow the same format as the behavioral evaluations in the main text (see Section 3.3 for the prompting method and Section 4.1 for results). Following the full input prompt, we appended one of the two following chain-of-thought templates:

1. **Model coordinate listing**: Step 1 lists the model-generated $(x, y)$ coordinates for all data points in the scatter plot. Step 2 states: "answer the question."

2. **Ground-truth coordinate listing**: Step 1 lists the ground-truth $(x, y)$ coordinates for all data points in the scatter plot. Step 2 states: "answer the question."

Following these chain-of-thought templates, we allow the model to generate a maximum of 1,000 new tokens with temperature set to 0. Examples of the two templates are displayed in Table 5.

**Prompt coordinate listing template**

```
STEP 1: Let's identify each point in the image
one by one.
- Red star: x=7, y=7
- Red circle: x=7, y=7
- Orange circle: x=6, y=7
- Green square: x=4, y=7
- Green triangle: x=3, y=5
- Blue square: x=1, y=3
- Blue triangle: x=1, y=3
- Orange star: x=6, y=1
STEP 2: **State the color and shape of the data
point with the smallest x-value**
BE CONCISE; ONLY RESPOND WITH THE ANSWER. The
color and shape of the data point with the
smallest x-value =
```

**Ground-truth coordinate listing template**

```
STEP 1: **Identify the (x, y) coordinates of all
points in the image.**
- Red star: x=5, y=6
- Red circle: x=7, y=7
- Orange circle: x=5, y=3
- Green square: x=4, y=7
- Green triangle: x=3, y=5
- Blue square: x=1, y=3
- Blue triangle: x=2, y=1
- Orange star: x=6, y=1
STEP 2: **State the color and shape of the data
point with the smallest x-value**
BE CONCISE; ONLY RESPOND WITH THE ANSWER. The
color and shape of the data point with the
smallest x-value =
```

**Task**: "Which data point has the largest y-value?"

Table 5: **Example of a prompt coordinate-listing template vs. a ground-truth coordinate listing template.** The image and task are displayed on the left, while the two templates are displayed on the right. The order of the objects listed is the same across templates for a given <task, image> pair. After the "=", the model is allowed to freely generate its response.

## C.4 DETAILED CHAIN-OF-THOUGHT PROMPTING RESULTS

This section contains results from Section 4.4 (Figure 4B) broken out by task for each model. See Figure 7 for results on FUGU tasks and Figure 8 for results on ensemble tasks.

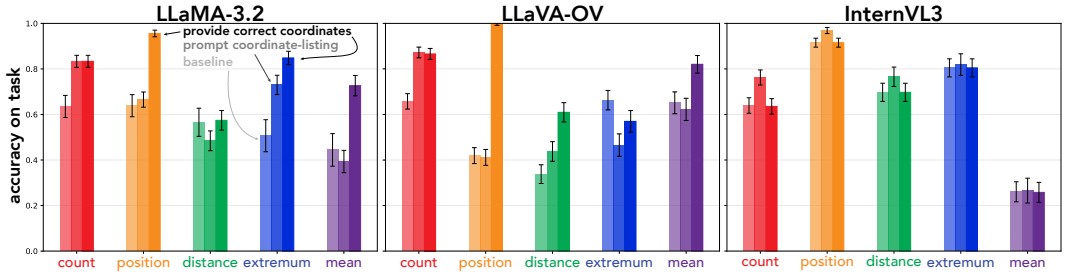

Figure 7: **Results from Section 4.4 broken out by task for FUGU tasks**.

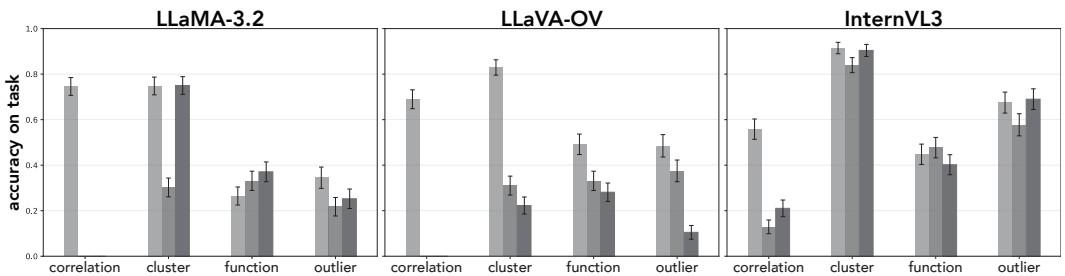

Figure 8: **Results from Section 4.4 broken out by task for ensemble tasks**.

## C.5 EXAMPLE MODEL RESPONSES

### C.5.1 LLAMA-3.2

| Task | Input Image | Prompt & Model Response | Correct? |
|------|-------------|-------------------------|----------|
| Count | 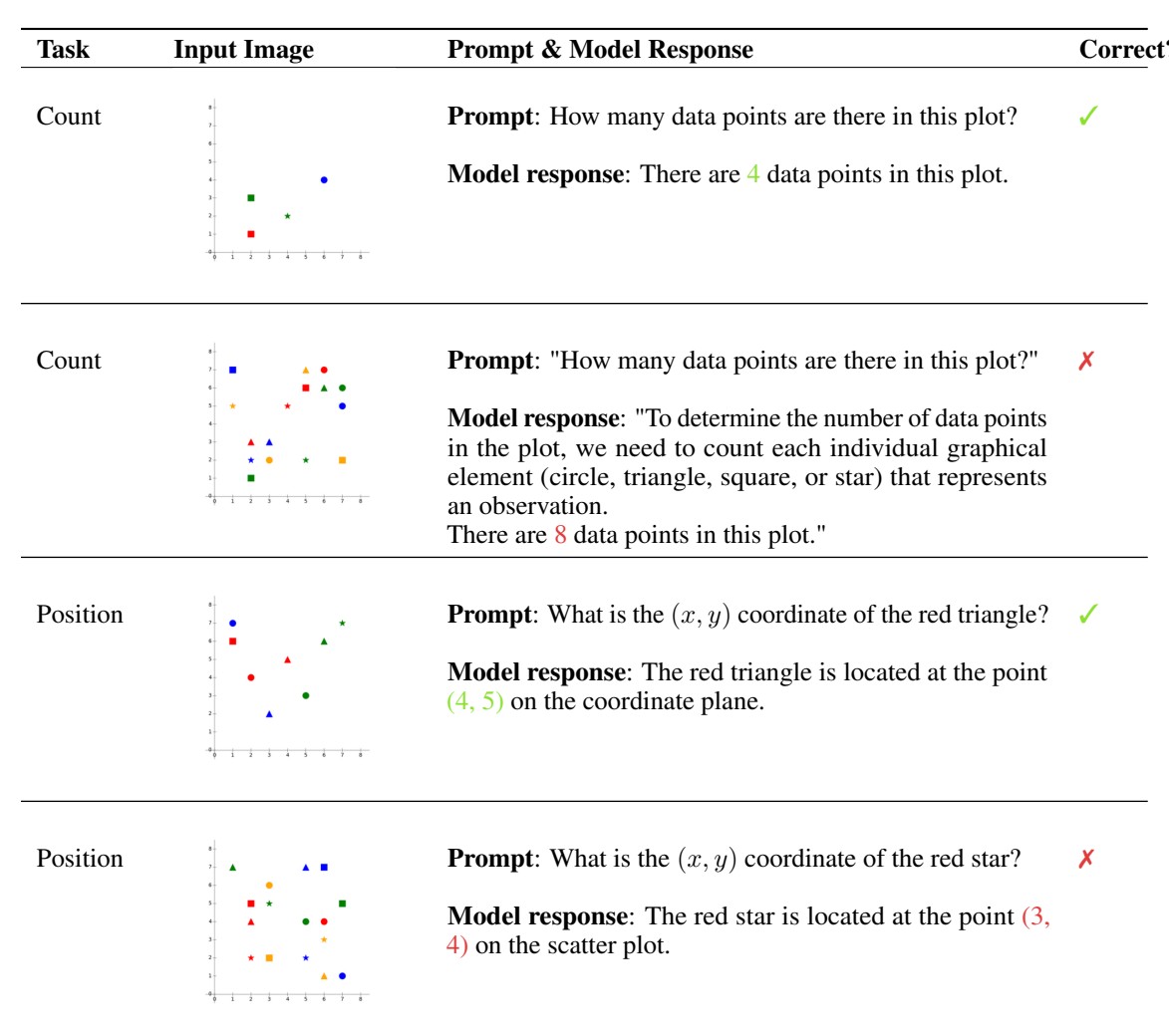 | **Prompt**: How many data points are there in this plot?

**Model response**: There are 4 data points in this plot. | ✓ |
| Count | | **Prompt**: "How many data points are there in this plot?"

**Model response**: "To determine the number of data points in the plot, we need to count each individual graphical element (circle, triangle, square, or star) that represents an observation.
There are 8 data points in this plot." | ✗ |
| Position | | **Prompt**: What is the $(x, y)$ coordinate of the red triangle?

**Model response**: The red triangle is located at the point $(4, 5)$ on the coordinate plane. | ✓ |
| Position | | **Prompt**: What is the $(x, y)$ coordinate of the red star?

**Model response**: The red star is located at the point $(3, 4)$ on the scatter plot. | ✗ |

Table 6: **Examples of full LLaMA-3.2 responses to the count and position tasks**.

| Task | Input Image | Prompt & Model Response | Correct? |
|------|-------------|-------------------------|----------|
| Distance | | **Prompt**: What is the distance between the red triangle and the blue square?

**Model response**: **Step 1: Identify the coordinates of the red triangle and the blue square.** The red triangle is located at the point (3, 6) on the coordinate plane. The blue square is located at the point (4, 2) on the coordinate plane.

**Step 2: Calculate the distance between the two points using the distance formula.** The distance formula is given by:

$$d = sqrt((x2 - x1)^2 + (y2 - y1)^2)$$

where d is the distance, (x1, y1) are the coordinates of the first point, and (x2, y2) are the coordinates of the second point.

Plugging in the values, we get:

$$d = sqrt((4 - 3)^2 + (2 - 6)^2)$$
$$= sqrt(1^2 + (-4)^2)$$
$$= sqrt(1 + 16)$$
$$= sqrt(17)$$

**Step 3: Round the distance to the nearest whole number.** Rounding sqrt(17) to the nearest whole number gives us 5. **Answer:** 5 | ✓ |
| Distance | | **Prompt**: What is the distance between the blue star and the green triangle?

**Model response**: **Step 1: Identify the coordinates of the blue star and the green triangle.** The blue star is located at (1, 2) and the green triangle is located at (7, 4).

**Step 2: Calculate the distance between the two points using the distance formula.** The distance formula is given by:

$$d = \sqrt{(x_2 - x_1)^2 + (y_2 - y_1)^2}$$

where $d$ is the distance, $(x_1, y_1)$ are the coordinates of the first point, and $(x_2, y_2)$ are the coordinates of the second point. Plugging in the values, we get:

$$d = \sqrt{(7 - 1)^2 + (4 - 2)^2} = \sqrt{6^2 + 2^2} = \sqrt{36 + 4}$$

$$= \sqrt{40} \approx 6.32$$

**Step 3: Round the distance to the nearest whole number.** Rounding 6.32 to the nearest whole number gives us 6. **Answer:** The distance between the blue star and the green triangle is approximately 6 units. | ✗ |

Table 7: **Examples of full LLaMA-3.2 responses to the distance task**.

| Task | Input Image | Prompt & Model Response | Correct? |
|------|-------------|-------------------------|----------|
| Minimum $x$ |  | **Prompt**: Which data point has the smallest x-value?

**Model response**: **Step 1: Identify the x-axis and y-axis.**
The x-axis is the horizontal axis, and the y-axis is the vertical axis.
**Step 2: Identify the data points.**
There are four data points in the scatter plot: a red circle, a green square, a blue triangle, and an orange star.
**Step 3: Determine the x-values of each data point.**
The x-values of the data points are as follows:
* Red circle: 3 * Green square: 7 * Blue triangle: 2 * Orange star: 4
**Step 4: Identify the data point with the smallest x-value.**
The data point with the smallest x-value is the blue triangle, which has an x-value of 2.
**Answer:** The data point with the smallest x-value is the blue triangle. | ✓ |
| Minimum $x$ |  | **Prompt**: Which data point has the smallest x-value?

**Model response**: **Step 1: Identify the x-axis values for each data point.**
* The x-axis values are: 0, 1, 2, 3, 4, 5, 6, 7, 8.
**Step 2: Identify the data point with the smallest x-value.**
* The data point with the smallest x-value is the one with the lowest x-axis value, which is 0.
**Step 3: Determine the color and shape of the data point with the smallest x-value.**
* The data point with the smallest x-value is a blue square.
**Answer:** The data point with the smallest x-value is a blue square. | ✗ |

Table 8: **Examples of full LLaMA-3.2 responses to minimum-$x$ extremum tasks**.

| Task | Input Image | Prompt & Model Response | Correct? |
|------|-------------|-------------------------|----------|
| Minimum $y$ |  | **Prompt**: Which data point has the smallest y-value? 

 **Model response**: **Step 1: Identify the data points in the scatter plot.** The scatter plot displays a total of 12 data points, each represented by a unique color and shape. The colors are red, green, blue, and orange, and the shapes are circles, triangles, squares, and stars. **Step 2: Determine the y-values of each data point.** The y-values of the data points are as follows: * Red circle: 7 * Red triangle: 7 * Red square: 5 * Red star: 2 * Green circle: 1 * Green triangle: 6 * Green square: 5 * Blue circle: 7 * Blue triangle: 4 * Blue square: 6 * Orange circle: 4 * Orange square: 5 **Step 3: Identify the data point with the smallest y-value.** The data point with the smallest y-value is the green circle, which has a y-value of 1. **Answer:** The data point with the smallest y-value is the green circle. | ✓ |
| Minimum $y$ |  | **Prompt**: Which data point has the smallest y-value? 

 **Model response**: **Step 1: Identify the data points in the scatter plot.** The scatter plot displays a total of 8 data points, each represented by a different color and shape. The colors are red, green, blue, and orange, and the shapes are circles, triangles, squares, and stars. **Step 2: Determine the y-values of each data point.** The y-values of each data point are as follows: * Red circle: 3 * Green square: 3 * Blue star: 4 * Orange triangle: 7 * Red triangle: 3 * Green circle: 4 * Blue square: 6 * Orange square: 7 **Step 3: Identify the data point with the smallest y-value.** The data point with the smallest y-value is the red triangle, which has a y-value of 3. **Answer:** The data point with the smallest y-value is the red triangle. ✗ | |

Table 9: **Examples of full LLaMA-3.2 responses to minimum-$y$ extremum tasks**.

| Task | Input Image | Prompt & Model Response | Correct? |
|---|---|---|---|
| Maximum $x$ |  | **Prompt**: Which data point has the largest x-value? 

 **Model response**: \*\*Step 1: Identify the x-axis.\*\* The x-axis is the horizontal axis, which represents the independent variable in the scatter plot. \*\*Step 2: Identify the data points.\*\* The data points are represented by different shapes (circles, triangles, squares, and stars) and colors (red, green, blue, and orange). \*\*Step 3: Determine the x-values of each data point.\*\* The x-values of each data point are as follows: \* Red circle: 1 \* Green square: 2 \* Blue triangle: 3 \* Orange star: 4 \* Red square: 5 \* Green triangle: 6 \* Blue circle: 7 \* Orange square: 8 \*\*Step 4: Identify the data point with the largest x-value.\*\* The data point with the largest x-value is the orange square, which has an x-value of 8. \*\*Answer:\*\* The data point with the largest x-value is the orange square. | ✓ |
| Maximum $x$ |  | **Prompt**: Which data point has the largest x-value? 

 **Model response**: \*\*Step 1: Identify the x-axis.\*\* The x-axis is the horizontal axis, which represents the independent variable in the scatter plot. \*\*Step 2: Identify the data points.\*\* The data points are represented by different shapes (circles, triangles, squares, and stars) and colors (red, green, blue, and orange). \*\*Step 3: Determine the x-values of each data point.\*\* The x-values of each data point are as follows: \* Red circle: 1.6 \* Green star: 7 \* Blue square: 2.8 \* Orange triangle: 3.2 \* Red star: 1.2 \* Green triangle: 6.4 \* Blue circle: 4 \* Orange square: 7.2 \*\*Step 4: Identify the data point with the largest x-value.\*\* The data point with the largest x-value is the orange square, which has an x-value of 7.2. \*\*Answer:\*\* The data point with the largest x-value is the orange square. | ✗ |

Table 10: **Examples of full LLaMA-3.2 responses to maximum-$x$ extremum tasks**.

| Task | Input Image | Prompt & Model Response | Correct? |
|------|-------------|-------------------------|----------|
| Maximum $y$ |  | **Prompt**: Which data point has the largest y-value?

**Model response**: **Step 1: Identify the data points in the scatter plot.**
The scatter plot displays a total of 8 data points, each represented by a different color and shape. The colors are red, green, blue, and orange, and the shapes are circles, triangles, squares, and stars.
**Step 2: Determine the y-values of each data point.**
The y-values of each data point are as follows:
* Red circle: 6.5 * Green circle: 4.5 * Blue circle: 5.5 * Orange circle: 7.5 * Red triangle: 5.5 * Green triangle: 4.5 * Blue triangle: 3.5 * Orange triangle: 7.5
**Step 3: Identify the data point with the largest y-value.**
The data point with the largest y-value is the orange square, which has a y-value of 7.5.
**Answer:** The data point with the largest y-value is the orange square. | ✓ |
| Maximum $y$ |  | **Prompt**: Which data point has the largest y-value?

**Model response**: **Step 1: Identify the data points in the scatter plot.**
The scatter plot displays 12 data points, each represented by a different color and shape (circle, triangle, square, or star). The colors are red, green, blue, and orange.
**Step 2: Determine the y-values for each data point.**
The y-values for each data point are as follows:
* Red circle: 6.5 * Red triangle: 4.5 * Red square: 3.5 * Red star: 7.5 * Green circle: 2.5 * Green triangle: 1.5 * Green square: 4.5 * Green star: 2.5 * Blue circle: 4.5 * Blue triangle: 3.5 * Blue square: 2.5 * Orange circle: 1.5
**Step 3: Identify the data point with the largest y-value.**
The data point with the largest y-value is the red star, with a y-value of 7.5.
**Answer:** The data point with the largest y-value is the red star. | ✗ |

Table 11: **Examples of full LLaMA-3.2 responses to maximum-$y$ extremum tasks**.

| Task | Input Image | Prompt & Model Response | Correct? |
|------|-------------|-------------------------|----------|
| Mean |  | **Prompt**: What is the (x, y) coordinate that is closest to the centroid, or arithmetic mean of the positions of all data points? 

 **Model response**: **Step 1: Identify the data points and their coordinates.** 
 The data points are represented by different colors and shapes, each with a unique (x, y) coordinate. 
 * Red triangle: (2.5, 3.5) * Orange square: (3, 3) * Green square: (2, 7) * Blue square: (6, 6) * Red star: (1.5, 1.5) 
 **Step 2: Calculate the centroid (arithmetic mean) of the x-coordinates and y-coordinates separately.** 
 To find the centroid, we need to calculate the average of all x-coordinates and the average of all y-coordinates. 
 * Average x-coordinate = (2.5 + 3 + 2 + 6 + 1.5) / 5 = 15 / 5 = 3 * Average y-coordinate = (3.5 + 3 + 7 + 6 + 1.5) / 5 = 21 / 5 = 4.2 
 **Step 3: Round the centroid coordinates to the nearest whole number.** 
 Rounded x-coordinate = 3 Rounded y-coordinate = 4 
 **Answer:** The (x, y) coordinate closest to the centroid is (3, 4). | ✓ |
| Mean |  | **Prompt**: What is the (x, y) coordinate that is closest to the centroid, or arithmetic mean of the positions of all data points? 

 **Model response**: **Step 1: Identify the data points in the scatter plot.** 
 The scatter plot displays four data points, each represented by a different color: red, green, blue, and orange. The x-values for these points are 2, 3, 5, and 7, respectively. The corresponding y-values are 7, 7, 6, and 7. 
 **Step 2: Calculate the mean of the x-values.** 
 To find the mean of the x-values, we add up all the x-values and divide by the total number of data points. 
 Mean x = (2 + 3 + 5 + 7) / 4 = 17 / 4 = 4.25 
 **Step 3: Calculate the mean of the y-values.** 
 To find the mean of the y-values, we add up all the y-values and divide by the total number of data points. 
 Mean y = (7 + 7 + 6 + 7) / 4 = 27 / 4 = 6.75 
 **Step 4: Determine the closest data point to the centroid.** 
 The centroid is the point (4.25, 6.75). We need to find the data point that is closest to this point. 
 The closest data point to the centroid is the green point at (3, 7). 
 **Step 5: Round the x-value and y-value to the nearest whole number.** 
 Rounded x-value = 3 Rounded y-value = 7 
 **Answer:** The (x, y) coordinate closest to the centroid is (3, 7). | ✗ |

Table 12: **Examples of full LLaMA-3.2 responses to mean task.**

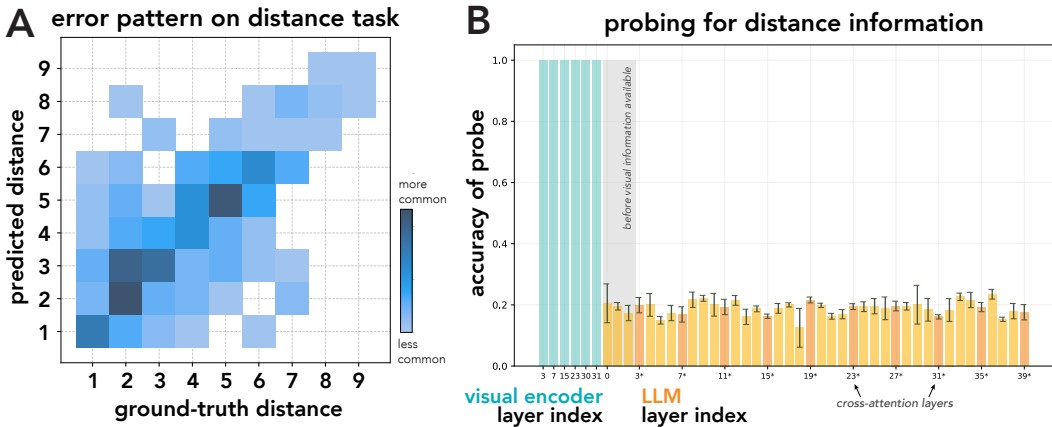

Figure 9: **Examining model failure on the distance task. A) LLaMA-3.2 11B's error patterns on the distance task.** Ground truth distances for all stimuli ($x$-axis) are plotted against model predicted distances for the same stimuli ($y$-axis). Darker hues represent that a given (ground truth, predicted distance) pair is more common. For larger distances, model predictions are typically closer to ground truth. For shorter distances, model predictions demonstrate much greater variance, although the bulk of distance predictions are accurate or close to accurate. **B) Probe test accuracy for distance in vision vs. language layers.**

## D FINE-TUNING DETAILS

We fine-tuned all parameters in each model. We conducted hyperparameter sweeps for each of the three models across learning rates $\in \{$1e-3, 1e-4, 1e-5$\}$ learning rate schedulers $\in \{$LinearLR, ReduceLROnPlateau, CosineAnnealingLR$\}$, and optimizers $\in \{$Adam, AdamW, SGD$\}$. For each model and dataset size combination, we selected the best-performing configuration based on held-out validation performance. All models achieved 98%-99% training accuracy, indicating successful optimization on the training objectives.

To generate sufficient unique images for the training datasets, we introduced additional variation beyond our original controlled settings: dot sizes, number of axis ticks, and numeric scales of tick labels. The "original" FUGU values of these parameters were included. The ensemble task stimuli vary in dot size, amount of noise, and distribution parameters (e.g. locations of the clusters, constants in the ground-truth functional relationship). The training datasets were roughly evenly distributed across all 9 tasks (FUGU + ensemble).

## E ERROR ANALYSIS

### E.1 ERROR CASE STUDY: LLaMA-3.2 ON THE DISTANCE TASK

In Section 4.4, we observed that LLaMA-3.2 11B's performance on the distance task did not improve over baseline when provided with the ground-truth $(x, y)$ coordinates of the data points. These results pointed to an additional bottleneck beyond coordinate extraction on the distance task, which we examine in greater detail in this section.

We found that predictions generally approximate the correct distances, typically deviating by only $\pm 1$ unit (Figure 9A). We observed higher variability in prediction accuracy for shorter distances, with the model occasionally producing substantial errors (predictions of 5-6 units for actual distances of 1 unit). Analysis of the model's reasoning reveals a consistent strategy: the model extracts coordinates for the relevant data points and applies the Euclidean distance formula to the extracted coordinates. Since predicted distances are frequently close to ground-truth, it is likely that arithmetic complexities introduced by the squaring and square root calculations in the Euclidean distance formula contribute significantly to inaccuracies, rather than coordinate extraction alone.

To what extent do the visual representations contribute to model failures on the distance task? To answer this question, we employed linear probes to predict distances between specific data point pairs from the model's vision or language representations. We found a stark disparity between vision and language processing on the distance task: probes trained on visual features achieved perfect 100% test accuracy, while those trained on language features failed to predict distances effectively despite cross-attention with visual features (Figure 9B). This finding provides insight into the model's processing limitations: despite the vision encoder linearly representing distance information, this information fails to linearly transfer into the LM, highlighting a potential limitation of how this architecture achieves the hand-off between the vision and language model components.

## E.2    ERROR PATTERNS

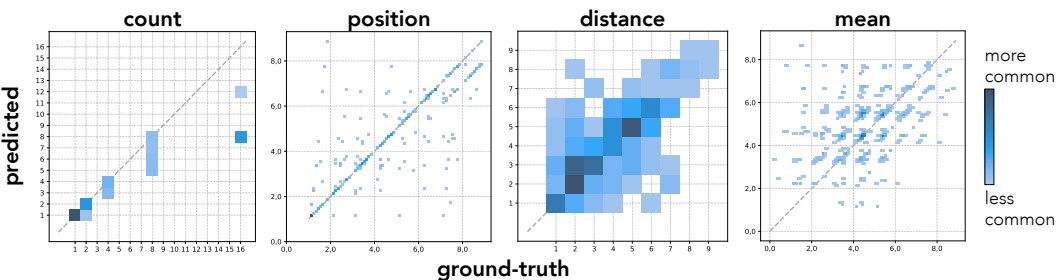

Figure 10: **Error patterns for LLaMA-3.2.**

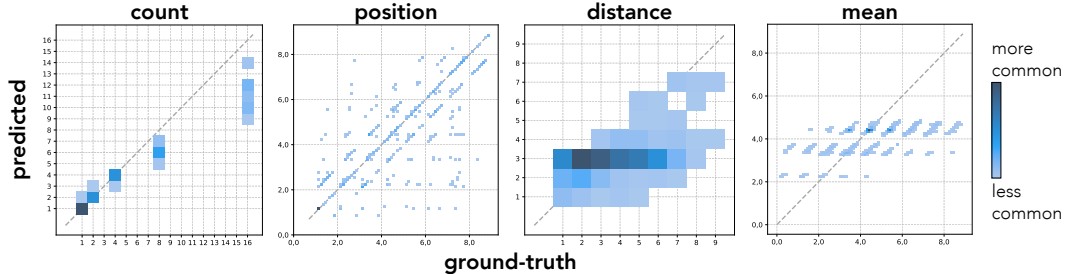

Figure 11: **Error patterns for LLaVA-OneVision.**

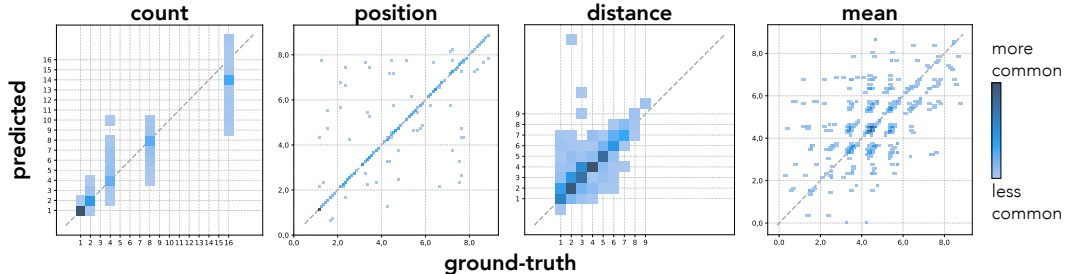

Figure 12: **Error patterns for InternVL3.**

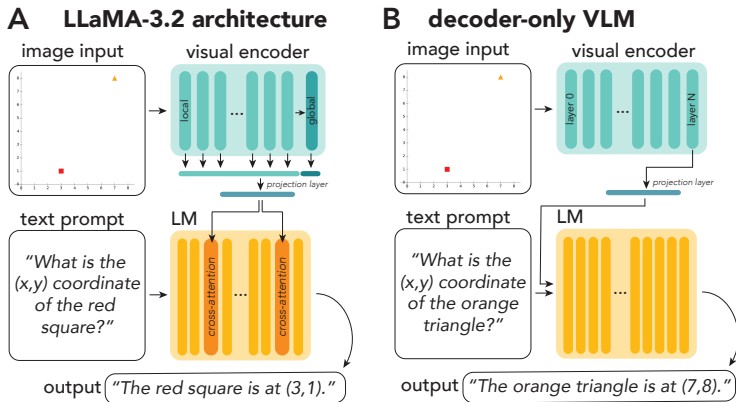

Figure 13: **Illustrations of the model architectures tested in the paper**. **A) LLaMA-3.2**. This model is a cross-attention based architecture in which visual features are first extracted from multiple layers of the vision encoder, concatenated, projected onto the dimension of the LM, and then used as the keys and values in cross-attention layers evenly spaced through the LM. **Decoder-only VLM architecture**. This diagram encompasses both LLaVA-OneVision and InternVL3. Vision features are extracted from the final layer of the vision encoder, projected into the dimension of the LM, and given to the LM as additional tokens in its input context.

## F  LINEAR PROBE DETAILS

### F.1  EXTRACTING VISION AND LANGUAGE FEATURES

We trained probes on individual layers from the vision encoder or the LLM. For each layer, vision features were collected by concatenating the 1,280-dimensional representations of all 1,601 image patches in that layer into a $(1,280 \times 1,601)$-dimensional vector. Language features were obtained by passing both an input image and the task prompt through the model, concatenating the 4,096-dimensional representations of each token in the task prompt, resulting in a $(\text{len}(\text{prompt}) \times 4,096)$-dimensional vector.

### F.2  FORMAT OF PROBE OUTPUTS

When probing for the position task, we encoded coordinates as 16-dimensional vectors, where the first 8 entries encode the $x$-coordinate and the latter 8 entries encode the $y$-coordinate. When probing for distance, the probe outputs are simply a length-9 one-hot vector (since 9 is the maximal distance).

## G  MODEL DETAILS

In this section, we provide additional details about the three model architectures tested in the main body of the paper: LLaMA-3.2B 11B (Grattafiori et al., 2024), LLaVA-OneVision 7B (Li et al., 2024), and InternVL3 14B (Zhu et al., 2025). See Figure 13 for diagrams of each model's architecture.

### G.1  LLAMA-3.2 11B DETAILS

LLaMA-3.2B 11B's architecture consists of three major components: the visual encoder, the language module (LM), and the vision-language adapter (Figure 13). The visual encoder consists of a standard Vision Transformer (ViT; Dosovitskiy et al. 2020) operating over image patches of 14x14 pixels, creating 40x40 image patches for an image of 560x560px. The ViT consists of a 14x14x3 convolutional filter to embed the image patches, then 32 standard vision transformer layers, followed by 8 gated self-attention layers. To feed visual information into the LM, the outputs of the 4th, 8th, 16th, 24th, 31st and final layers are concatenated along the feature dimension for each image patch. These features are projected into the same dimensionality as the LM and are then used to construct

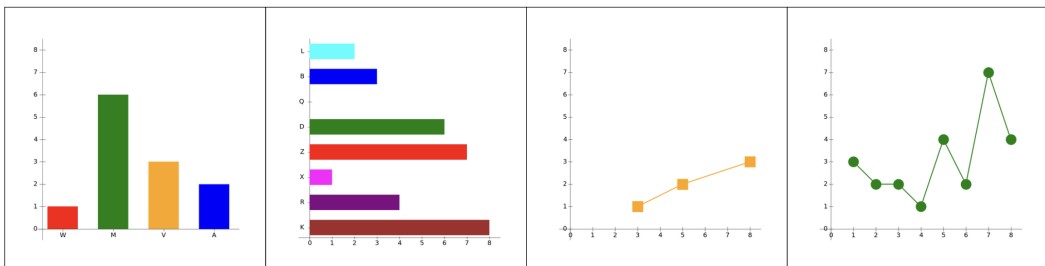

Figure 14: **Example FUGU bar and line charts**. Each of the four charts is an individual stimulus.

the keys and values in cross-attention layers in the LM (Alayrac et al., 2022). The LM consists of a decoder only transformer in which a cross-attention layer occurs every 4th layer. The cross-attention layer uses queries from the LM and keys and values from the vision representations. When combining the vision and language data modalities during training, the LM's parameters are frozen whereas the vision encoder and cross attention layers are not (Grattafiori et al., 2024).

### G.2 LLaVA-OneVision 7B details

See (Li et al., 2024).

### G.3 InternVL3 14B details

See (Chen et al., 2024).

## H Generalizing to bar and line charts

In this section, we reproduce sections 4.1, 4.3, 4.4, and 4.5 on two additional chart types: bar charts and line charts.

### H.1 Bar and line chart dataset details

We generated two separate datasets, one for each new chart type. We refer to these datasets as FUGU-Bar and FUGU-Line. Both datasets contained $1,000$ <task, image> pairs, balanced across number of data points and task type.

#### H.1.1 Dataset generation

As in Section 3.1, we procedurally generated bar and line charts. The charts contain $n \in \{2, 3, 4, 8\}$ bars or data points. To ensure that prompts can refer to particular bars in the bar charts without ambiguity, each bar in a given chart was rendered with a unique color drawn randomly from 8 possibilities (red, green, blue, orange, purple, brown, cyan, magenta). Each line chart contains a single line of a uniform, randomly chosen color, and the $(x, y)$-points are represented by a marker of a single shape (drawn from four possible shapes: square, circle, triangle, and star). The plots were rendered with a white background and black axes, with integer tick marks and labels ranging from 0 to 8 on numeric axes. Line charts were rendered with two numeric axes; bar charts were rendered with numeric "value" axes and categorical $x$-axes, where each bar is labeled by a unique, randomly chosen alphabetical letter (e.g. "G"). Bar charts were generated in both horizontal and vertical configurations. Individual data points were randomly sampled to generate each plot. See Figure 14 for some example charts.

#### H.1.2 Tasks

We adapted the five basic FUGU tasks—count, position, distance, extremum, and mean—for the new bar and line chart stimuli. The prompts for each task were kept as similar as possible to the prompts

used for scatter plots, with slight adjustments to the wording as needed. Example prompts are given in Table 13.

| Task | Bar prompt | Line prompt |
|---|---|---|
| Count | "How many bars are shown in this chart?" | "How many markers (data points) are plotted in this line chart?" |
| Position | "What is the value (height) of the red bar labeled 'Q'?" | "What is the $y$-value of the data point at $x = 5$?" |
| Distance | "What is the difference in value between the red bar (category 'Q') and the green bar (category 'B'), rounded to the nearest whole number?" | "What is the difference in $y$-values between the data point at $x = 5$ and the data point at $x = 2$?" |
| Extremum | "Which bar has the smallest \| largest $\{x \mid y\}$-axis value? Identify it by its color." | "What is the smallest \| largest $\{x \mid y\}$-value of the data points?" |
| Mean | "What is the average (mean) value across all bars?" | "What is the average $y$-value of all data points?" |

Table 13: **Example prompts for bar and line charts.**

## H.2 BAR AND LINE CHART RESULTS

In this section, we reproduce Sections 4.1 (behavioral evaluation), 4.3 (point listing analysis), 4.4 (chain-of-thought interventions), and 4.5 (linear probes) on FUGU-Bar and FUGU-Line using the three main models (LLaMA-3.2, LLaVA-OneVision, InternVL3).

### H.2.1 BEHAVIORAL EVALUATION

We report task performance averaged across tasks as well as for each task for each of the models in Table 14. We also break down accuracy vs. the number of data points for each chart type in Table 15. As in Section 4.1, model performance degrades as the number of points in the plots increases.

FUGU-Bar

| Model | Count | Position | Distance | Extremum | Mean | Avg. |
|---|---|---|---|---|---|---|
| LLaMA-3.2 | 100.0 | 96.2 | 100.0 | 95.0 | 91.8 | 96.6 |
| LLaVA-OV | 90.6 | 98.1 | 75.0 | 40.0 | 75.5 | 75.8 |
| InternVL3 | 100.0 | 100.0 | 100.0 | 94.0 | 91.8 | 97.2 |

FUGU-Line

| Model | Count | Position | Distance | Extremum | Mean | Avg. |
|---|---|---|---|---|---|---|
| LLaMA-3.2 | 59.0 | 79.0 | 78.0 | 70.0 | 60.0 | 69.2 |
| LLaVA-OV | 81.0 | 70.0 | 47.0 | 66.5 | 56.0 | 63.8 |
| InternVL3 | 59.0 | 99.0 | 97.0 | 89.5 | 96.0 | 88.1 |

Table 14: **Task accuracy on FUGU-Bar and FUGU-Line.**

### H.2.2 POINT LISTING ANALYSIS

In Section 4.3, we observed that models often spontaneously list $(x, y)$-coordinates in their chain-of-thought in order to solve the original FUGU tasks. We find that this behavior generalizes to bar and line charts, and that for LLaMA-3.2 and LLaVA-OneVision, point listing accuracy noticeably degrades as plot complexity (i.e. the number of points) increases (see Figure 15). InternVL3, however, maintains high point-listing accuracy even as the number of points increases—this is also consistent with this model's behavior on scatter plots with a sample size of 8 or less (Figure 6).

| FUGU-Bar | | | | |
|---|---|---|---|---|
| **Model** | **n=2** | **n=3** | **n=4** | **n=8** |
| LLaMA-3.2 | 100.0 | 98.9 | 97.7 | 91.2 |
| LLaVA-OV | 97.2 | 78.0 | 67.0 | 54.9 |
| InternVL3 | 100.0 | 100.0 | 97.7 | 91.2 |

| FUGU-Line | | | | |
|---|---|---|---|---|
| **Model** | **n=2** | **n=3** | **n=4** | **n=8** |
| LLaMA-3.2 | 80.0 | 64.4 | 66.5 | 66.0 |
| LLaVA-OV | 84.0 | 59.9 | 60.2 | 65.1 |
| InternVL3 | 100.0 | 96.0 | 82.4 | 81.4 |

Table 15: **Task accuracy on FUGU-Bar and FUGU-Line degrades as the number of points increases.** This mirrors results in Figure 2.

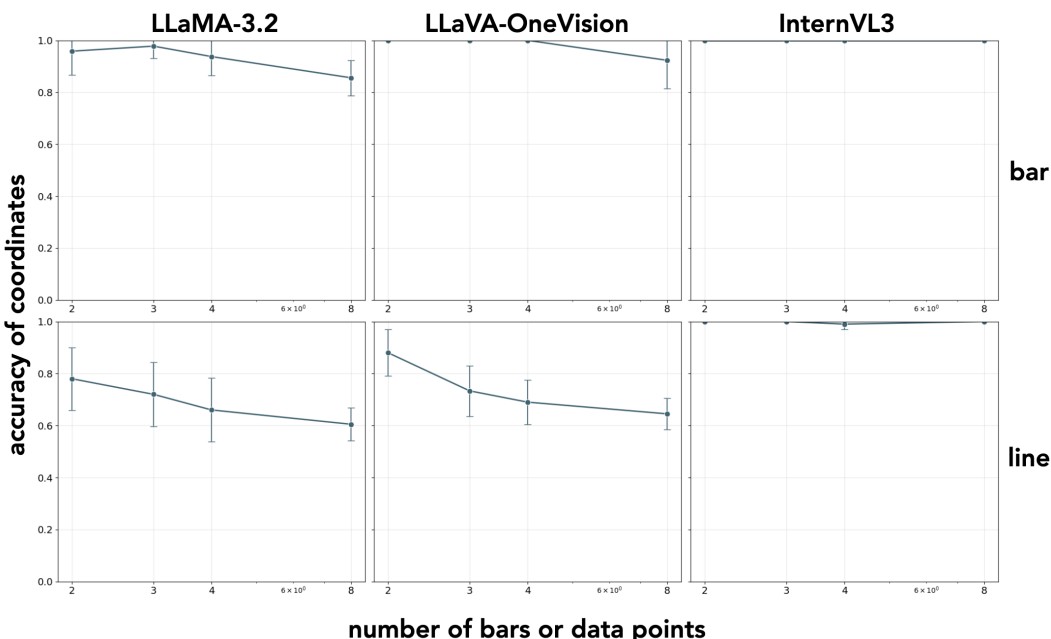

Figure 15: **Accuracy of listed coordinates on bar charts (top row) and line charts (bottom row).**

### H.2.3 CHAIN-OF-THOUGHT INTERVENTIONS

In Section 4.4, we found that providing models with ground truth scatter points in the LM's context improved model performance on FUGU tasks; this confirmed that accurate point listing is, indeed, a bottleneck for models on FUGU. Does this result also hold for bar and line charts? We replicated the same analysis described in Section 4.4 for FUGU-Bar and FUGU-Line. We found that providing ground truth point lists significantly improved performance for LLaVA-OneVision on both bar and line charts and for LLaMA-3.2 on line charts (see Figure 16). Ground truth point lists do not impact model performance for InternVL3 for either chart type, which already exhibited very high accuracy—nor does it impact performance for LLaMA-3.2 on bar charts, which was already near ceiling performance at baseline. The pattern observed on scatter plots generally holds for <model, chart> combinations where baseline performance is relatively low, suggesting that point listing is indeed a bottleneck on these chart types as well.

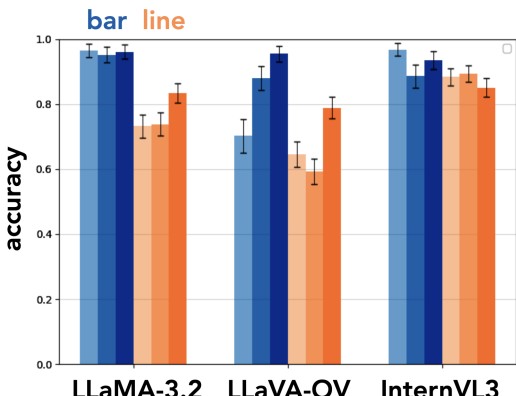

Figure 16: **Providing ground truth points in the chain-of-thought often improves performance on `FUGU-Bar` and `FUGU-Line`.** The blue "bar" bars show accuracy averaged across the five `FUGU-Bar` tasks, while the orange "line" bars show accuracy averaged across the five `FUGU-Line` tasks. The middle bars show accuracy when the model is provided with its own generated coordinates for each data point in the scatter plot as part of its chain-of-thought, while the darkest bars show accuracy when models are given ground-truth coordinates. The performance gains offered by the ground-truth listing on tasks where baseline performance is relatively low suggests that accurate coordinate extraction is a significant bottleneck in these settings.

| FUGU-Bar | | |
|---|---|---|
| **Model** | **vision encoder (avg.)** | **LM (avg.)** |
| LLaMA-3.2 | 99.9 | 19.3 |
| LLaVA-OV | 100.0 | 83.5 |
| InternVL3 | 100.0 | 82.7 |

| FUGU-Line | | |
|---|---|---|
| **Model** | **vision encoder (avg.)** | **LM (avg.)** |
| LLaMA-3.2 | 100.0 | 19.4 |
| LLaVA-OV | 99.8 | 72.0 |
| InternVL3 | 100.0 | 55.0 |

Table 16: **Probe test accuracy on bar and line charts, averaged across vision and language layers.**

### H.2.4 LINEAR PROBES

In Section 4.5, we found that $(x, y)$-coordinate information is perfectly decodable from vision encoder representations using linear probes. After information moves into the LM, probe accuracy drops, suggesting that the information is no longer linearly decodable (see Figure 5). Overall, this pointed to a degradation of task-relevant information that occurs as a result of the vision-language handoff. Does this result hold for bar and line charts?

Using the same methodology described in Section 3.5, we train linear probes to decode either bar values (for `FUGU-Bar`) or $y$-values of data-points (for `FUGU-Line`). For bar charts, we train probes to predict the value of a bar of a given color; in the training set, this target bar appears in all possible $x$ and $y$ positions, but a subset of $x$, $y$ combinations are held out for testing. For line charts, we train probes to predict the $y$-value of a data-point with a special marker that has a different shape and color from the other markers in the chart. Similarly to the bar charts, the target data point is seen at all $x$ and $y$ positions during training but not all $(x, y)$ combinations—these are held out for testing.

We report probe accuracy averaged across vision and language layers for each of the three main models in Table 16. We find a pattern of results that recapitulates our findings on scatter plots in Section 4.5: values are nearly perfectly linearly decodable from visual representations, but probe

accuracy drops off for language representations. This suggests that the vision-language handoff issue we identified on scatter plots generalizes to bar and line charts. Note that the drop-off is particularly severe for LLaMA-3.2—this is likely because the average includes layers before the first infusion of visual information, which have near zero probe accuracy.

# I DOES POINT LISTING GENERALIZE TO REAL PLOTS?

In Section 4.3, we observed that models we observed that models often spontaneously list $(x, y)$-coordinates in their chain-of-thought in order to solve FUGU tasks. As a consequence, errors in point listing propagate to downstream reasoning. Furthermore, in Section 4.4, we found that accurate coordinate listing represents a significant bottleneck for models. However, it is possible that this coordinate listing behavior is specific to synthetic FUGU tasks. Do models exhibit similar behavior on real data visualizations? If so, this suggests that a similar bottleneck could be encountered on real-world data, indicating that the insights gleaned from FUGU can be directly applied to model behavior "in the wild."

We evaluated our three main models (LLaMA-3.2, LLaVA-OneVision, InternVL3) on CharXiv (Wang et al., 2024), a dataset of real plots scraped from scientific papers on arXiv. We analyzed each model's chain-of-thought on the CharXiv items and counted the frequency of point listing behavior using the same method we used for evaluation (two LLM judges + human validated subset)—that is, how many times models attempt to list the values of data-points from the data visualization as part of their solution to a question. We found that LLaMA-3.2 exhibited point listing behaviors on around $20\%$ of the items, while InternVL listed points for around $15\%$. This indicates that point listing is a common behavior that also emerges on real data, and that our findings may be relevant for improving performance on real data visualizations.

