# OpenReview forum: "Diagnosing Bottlenecks in Data Visualization Understanding by Vision-Language Models"
_ICLR.cc/2026/Conference — Submitted to ICLR 2026_

### Official Review · Reviewer_oyTB · 2025-10-29

**Soundness:** 3
**Presentation:** 3
**Contribution:** 2
**Rating:** 4
**Confidence:** 4

**Summary:**

This paper seeks to investigate the bottleneck in Vision-Language Models (VLMs) for data visualization understanding. To this end, the authors introduce FUGU, a data visualization understanding benchmark consists of synthetic scatter plots and 9 tasks (5 basic and 4 ensemble). They evaluate three modern VLMs with mechanistic interpretability techniques (activation patching and linear probes). Based on the evaluation results, they argue that the bottleneck is caused by the vision-language handoff.

**Strengths:**

•	The flexibility and granularity (controlled visual attributes, and variable plot complexity) of FUGU enable precise probing of model weaknesses. The experiments are thorough, comparing three contemporary VLMs and systematically evaluating their performance with different visualization complexity and tasks.
•	The paper is well written and clearly introduces the motivation and experimental findings.

**Weaknesses:**

- Limited generalizability of the dataset
Because FUGU comprises only scatter plots, the generalizability of the paper’s conclusions is limited. Real-world data visualizations are more complex and diverse, incorporating various chart types (e.g., bar, line, pie charts), dense textual annotations, legends, and visual noise. The bottlenecks identified in this study may be specific to FUGU. It is important to evaluate whether the bottlenecks vary by visualization types and scale with both visualization complexity (from simple charts to rich infographics) and question complexity (from basic queries to complex data insight queries such as those in ChartQAPro).

-  The nature of the identified bottleneck seems task-dependent
The claim of the VLM architectural bottleneck is weakened by the fact that this bottleneck task-dependent. In Section 4.6, one experiment demonstrates that providing ground-truth coordinates harms performance on ensemble tasks. This suggests that the language part of VLMs probably faces a bottleneck in this context. Thus, the bottleneck is not a fixed architectural limitation but a task-specific capacity mismatch between model components and task demands.  This weakens the claim of a fundamental architectural limitation and shifts the contribution from identifying a fundamental architectural constraint to characterizing a capacity limitation in current VLMs. This is a more modest result than the paper suggests.

- Results on fine-tuning do not fully support the claim
In Sections 3.1 and 4.7, the authors claim that FUGU is extremely difficult for current VLMs and cannot be fully solved directly through fine-tuning. However, the fine-tuned InternVL3 14B performs very well on simple tasks (count 100, position 99.2). This suggests, first, that the bottleneck may shift after fine-tuning. Second, if a 14B model can solve simple tasks with fine-tuning, larger fine-tuned models may also handle additional FUGU tasks.

**Questions:**

- Why are the values of Gemini 2.5 Pro in Tables 2 and 3 missing?
- Considering the good performance of Gemini 2.5 Flash in Tables 2 and 3, is FUGU not difficult for Gemini 2.5 Pro?
- line 150: VLMS -> VLMs

---

> ### Author Response · Authors · 2025-12-03
> **Response to Reviewer oyTB**
>
> We thank the reviewer for their careful reading and valuable insights. Find a detailed point-by-point response below.
>
> ## Weaknesses
>
> ### Weakness 1: "Limited generalizability"
> We thank the reviewer for raising this point and agree that the work would be strengthened by including additional chart types. We have extended FUGU to include bar charts and line charts and reproduced all core analyses. Please see the global response.
>
> We additionally evaluate on CharXiv \[1\], a dataset of real scientific plots. We observed that LLaMA-3.2 and InternVL3 display point listing behaviors on about 20% and 15% of plots respectively. This indicates that this behavior emerges “in the wild,” suggesting that our analysis of point-listing failures in FUGU may apply more broadly. This evaluation has been added to the appendix (Appendix I).
>
> ### Weakness 2: "The nature of the identified bottleneck..."
> We appreciate the reviewer’s point and agree that we should clarify the scope of our claim. We do not argue that the vision-language handoff reflects a single, fundamental architectural limitation that explains all failures. Rather, across models and across many basic tasks, we observe a consistent and replicable point of degradation: although task-relevant visual information is well-represented in the vision encoder, it is not reliably accessed or used by the language module. We refer to this pattern as a “bottleneck” because it reflects suboptimal information flow between components, not because we claim it is the only source of difficulty or that it is immutable.
> Indeed, we identify multiple sources of failure:
> * A major subset of failures arise specifically from information lost at the vision-language interface.
> * Some failures are LM specific (e.g., distance-task errors due to arithmetic steps, Appendix E.1).
> * Some failures seem data-related, as suggested by the fine-tuning improvements (Section 4.7).
>
> Thus, our contribution is to characterize and localize one particularly important and recurrent point of breakdown, not to exclude the existence of others. We will review the paper to ensure that all of our claims are clear, scoped, and consistent with the above.
>
> 3. "Results on fine-tuning..."
> We thank the reviewer for raising this important point. The reviewer is correct that fine-tuning substantially improves InternVL3’s performance on several simple tasks (e.g., count and position). We agree that this indicates the bottleneck may not necessarily be a hard architectural ceiling, and we will clarify this in the camera-ready version of the paper.
>
> However, two observations limit the extent to which fine-tuning resolves the identified challenges:
> 1\. Fine-tuning does not close the gap to ceiling performance across all tasks or models. Despite aggressive hyperparameter sweeps and 100k training examples, no models reach overall ceiling performance (Table 1). LLaMA-3.2 and LLaVA-OneVision in particular still exhibit a 20-25 point gap. This suggests that there may be a key difference between InternVL3 and these other two architectures that accounts for their difference in performance.
> 2\. Even for InternVL, performance does not reach ceiling performance on harder tasks. InternVL3 is unusually strong throughout our experiments—including in coordinate extraction—but its performance still lags on multi-point ensemble tasks and does not reach ceiling after fine-tuning. This indicates that while fine-tuning can alleviate some difficulties, it does not eliminate the challenges identified in Sections 4.4-4.6.
> 3\. We aim to evaluate general-purpose models. Our emphasis is on understanding the zero-shot general multimodal reasoning ability of current VLMs. From this perspective, the bottleneck we identify is still highly relevant: even models with powerful vision encoders fail to reliably pass information to the LM without task-specific data augmentation.
> We will adjust Sections 3.1 and 4.7 to ensure that all claims are appropriately scoped and to clarify that our central claim concerns the current capabilities of general-purpose VLMs rather than the absolute limitations of fine-tuning.
>
> ## Questions
>
> > Why are the values of Gemini 2.5 Pro in Tables 2 and 3 missing?
>
> Thank you for catching this omission. The empty row for Gemini Pro reflects an incomplete evaluation that we intended to remove from the appendix prior to submission. We apologize for the oversight. We are currently finalizing the evaluation for this model and will include the complete results in the camera-ready version.
>
> > Considering the good performance of Gemini 2.5 Flash in Tables 2 and 3, is FUGU not difficult for Gemini 2.5 Pro?
>
> Gemini 2.5 Pro likely exhibits strong performance on FUGU!
>
> > line 150: VLMS > VLMs
>
> We thank the reviewer for noticing this and have corrected it.
>
> ## References
> \[1\] Wang, Z., Xia, M., He, L., Chen, H., Liu, Y., Zhu, R., ... & Chen, D. (2024). Charxiv: Charting gaps in realistic chart understanding in multimodal llms.

---

### Official Review · Reviewer_4DYu · 2025-10-30

**Soundness:** 3
**Presentation:** 2
**Contribution:** 3
**Rating:** 6
**Confidence:** 4

**Summary:**

The paper introduces FUGU, a controlled benchmark to diagnose where VLMs fail to understand basic charts. FUGU focuses on highly controlled, synthetic scatter plot tasks designed to probe models' abilities to extract and reason about quantitative information at multiple levels of complexity. The paper systematically evaluates three widely used VLM architectures (LLaMA-3.2, LLaVA-OneVision, InternVL3) using behavioral analysis, causal interventions (activation patching), and linear probes to track information flow and identify bottlenecks. The key findings implicate the hand-off between vision encoders and language modules as a major source of failure, rather than the underlying representation or reasoning capacities of either module in isolation.

**Strengths:**

+ The dataset design is simple, clear, and well-controlled, allowing for a clean analysis of specific model behaviors and error sources
+ The combination of multiple analysis methods provides different aspects to assess model behavior
+ The conclusion brought by linear probs experiments is interesting and convincing

**Weaknesses:**

- The dataset scope is narrow. FUGU focuses on scatterplots with limited data points, no occlusion, and fixed glyphs. Real charts usually include bars/lines, partial occlusion, diverse scaled axes, and additional legends/annotations. It remains unclear whether the experimental results and conclusions would still hold for other chart types or in-the-wild data.

**Questions:**

- How robust are the conclusions if the scatterplot contains more points, and the axes are non-integer?

---

> ### Author Response · Authors · 2025-12-03
> **Response to Reviewer 4DYu**
>
> Thank you for the thoughtful comments and feedback! Find a point-by-point reply below.
>
> ## Weaknesses
>
> ### Weakness 1: "The dataset scope is narrow"
> We thank the reviewer for raising this point and agree that the work would be strengthened by including additional chart types. We have taken a substantial step in this direction by extending FUGU to include two additional chart types—bar charts and line charts—and reproducing all core analyses (behavioral evaluation, coordinate/height extraction accuracy, chain-of-thought interventions, and linear probes) across the three main models. We find that the key patterns observed in scatter plots are largely recapitulated in these new settings, reinforcing our original conclusions. Full results can be found in a newly added appendix (Appendix H).
>
> We additionally evaluate on CharXiv \[1\], a dataset of real scientific plots. In particular, we counted how often models exhibit point listing behaviors in their chain-of-thought in order to solve these tasks. Since point listing represents the most significant bottleneck we identify, this behavior emerging on real data would suggest that the insights we uncover using FUGU may be relevant for real-world data visualizations. We observed that LLaMA-3.2 and InternVL3 display point listing behaviors on about 20% and 15% of plots respectively. This indicates that this behavior does indeed emerge “in the wild,” suggesting that our analysis of point-listing failures in FUGU may apply more broadly. This evaluation has been added to the appendix in the revised PDF (Appendix I).
>
> ## Questions
>
> ### Question 1
> > _"How robust are the conclusions if the scatterplot contains more points, and the axes are non-integer?"_
>
> We thank the reviewer for their question. Our results in Section 4.1 show that model performance reliably degrades as the number of points increases across tasks. To address the reviewer’s question about non-integer axes, we generated FUGU plots with uniformly distributed (non-integer) point locations, and we report these results in a new appendix section. The overall qualitative patterns remain unchanged, though performance is somewhat lower due to the added precision required in the extraction step. This further supports our conclusions.
>
> ## References
> \[1\] Wang, Z., Xia, M., He, L., Chen, H., Liu, Y., Zhu, R., ... & Chen, D. (2024). Charxiv: Charting gaps in realistic chart understanding in multimodal llms.

---

### Official Review · Reviewer_jg4B · 2025-10-31

**Soundness:** 2
**Presentation:** 2
**Contribution:** 2
**Rating:** 4
**Confidence:** 4

**Summary:**

This paper investigates why modern vision-language models (VLMs) fail to understand data visualizations, arguing it's unclear if the fault lies in the visual encoder, language module, or their interface. To diagnose this, the authors introduce FUGU, a new benchmark of fine-grained "unit tests" for chart-understanding capabilities like extracting data points or calculating statistics. Evaluating three VLMs (LLaMA-3.2, LLaVA-OneVision, InternVL3), they find poor performance that degrades rapidly as the number of data points increases. Using diagnostic tools like linear probes and activation patching, the authors pinpoint a key bottleneck. They find that while the visual encoder does successfully capture low-level information (like coordinates), this information is "scrambled" or lost at the vision-language connector and in the early layers of the language model. This core architectural flaw, rather than a failure of visual perception, is the primary source of error and persists even after fine-tuning. The work's primary limitations are its focus on synthetic scatter/bar charts and the computational cost of its diagnostic methods.

**Strengths:**

1. Introduces FUGU, a diagnostic benchmark designed to "unit test" the fine-grained capabilities of VLMs on data visualizations.
2. Provides a clear and localized diagnosis for VLM failures, identifying the vision-language connector and early LM layers as the primary bottleneck.
3. The work appears reproducible due to clear descriptions of the FUGU benchmark tasks and the diagnostic methods.

**Weaknesses:**

* The FUGU benchmark's scope is currently narrow, focusing on synthetic scatter plots (Sec 3.1). This makes it unclear if the findings and the identified bottleneck generalize to other common chart families (e.g., bar charts, line graphs, histograms) or to more complex, real-world visualizations with varied aesthetics, occlusions, or multiple panels.
* The paper's contribution is primarily diagnostic. While it successfully identifies the location of the information bottleneck (Sec 5.3), it does not proceed to propose or empirically test any architectural solutions or mitigation strategies to address this flaw.
* The evaluation relies on a judge prompt and regex-based checks for scoring (Sec 4.1). While the high agreement rate is noted, the lack of a human-rated validation subset with inter-rater agreement means the robustness of the automated scoring is not fully confirmed.
* The tasks within FUGU (e.g., counting, position, distance, mean) appear to rely heavily on a single core skill: accurately extracting all (x,y) coordinates. This might over-weight one specific failure mode rather than testing a diverse set of reasoning capabilities.

**Questions:**

N/A

---

> ### Author Response · Authors · 2025-12-03
> **Response to Reviewer jg4B**
>
> Thank you for your thoughtful review! Find a detailed point-by-point response below.
>
> ## Weaknesses
>
> ### Weakness 1: "The FUGU benchmark's scope is currently narrow..."
> We thank the reviewer for raising this point and agree that the work would be strengthened by including additional chart types. We have taken a substantial step in this direction by extending FUGU to include two additional chart types—bar charts and line charts—and reproducing all core analyses across the three main models. We find that the key patterns observed in scatter plots are largely recapitulated in these new settings, reinforcing our original conclusions. Full results can be found in a newly added appendix (Appendix H).
>
> We additionally evaluate on CharXiv \[1\], a dataset of real scientific plots. In particular, we counted how often models exhibit point listing behaviors in their chain-of-thought in order to solve these tasks. Since point listing represents the most significant bottleneck we identify, this behavior emerging on real data would suggest that the insights we uncover using FUGU may be relevant for real-world data visualizations. We observed that LLaMA-3.2 and InternVL3 display point listing behaviors on about 20% and 15% of plots respectively. This indicates that this behavior does indeed emerge “in the wild,” suggesting that our analysis of point-listing failures in FUGU may apply more broadly. This evaluation has been added to the appendix in the revised PDF (Appendix I).
>
> ### Weakness 2: "The paper's contribution is primarily diagnostic."
>
> While we reserve an empirical investigation of model results for future work, we have greatly expanded the discussion to articulate concrete architectural directions suggested by our findings:
>
> * Connectors for spatial reasoning. Current multimodal connectors are trained on image-caption data and may not learn mappings that preserve fine-grained spatial structure. Connectors explicitly optimized to extract and represent spatial relations could mitigate this.
> * Dynamic, prompt-conditioned connectors. Architectures that adapt the visual representation to the task described in the prompt may reduce information loss during the handoff.
> * Reasoning with generative vision models. Instead of forcing all reasoning to occur in language space, models could also learn reason in a rich visual latent space, leveraging recent advances in generative image & video modeling. Recent work has found that video models are capable of solving a variety of reasoning problems zero shot by generating intermediate frames rather than language tokens \[2, 3\]. Developing this approach would avoid the issue of lossy vision-to-language mappings entirely.
>
> ### Weakness 3: "The evaluation relies on a judge prompt..."
> We thank the reviewer for this suggestion. We did conduct human validation study but omitted it. Human raters exhibited very high agreement with the automated judge on a randomly sampled subset of 50 items per model (50/50 adjudicated correctly). This supports the robustness of our scoring protocol.
>
> 4. The tasks within FUGU (e.g., counting, position, distance, mean) appear to rely heavily on a single core skill: accurately extracting all (x,y) coordinates. This might over-weight one specific failure mode rather than testing a diverse set of reasoning capabilities.
>
> This is an important point and we appreciate the opportunity to clarify the rationale for key decisions we made in designing FUGU and FUGU-ensemble. The task suite was designed to be hierarchically organized, with more advanced tasks (e.g., estimating the strength of the correlation between two variables; assigning a data point to a cluster) building on core competencies shared with the more basic tasks (i.e., reporting the (x,y) position of a specific data point). The hierarchical nature of the task suite is important because it helps with diagnosing where models break down in this hierarchy of competencies.
>
> It did not have to be the case that models consistently struggled with extracting (x,y) coordinates in the base FUGU tasks. Instead, it could have been that errors made by the model were predominantly due to errors in mathematical operations and/or statistical inference performed downstream of coordinate extraction. As such, we find our findings to be somewhat striking—that current VLMs consistently struggle with such a foundational component of many data visualization understanding tasks.
>
> ### References
> \[1\] Wang, Z., Xia, M., He, L., Chen, H., Liu, Y., Zhu, R., ... & Chen, D. (2024). Charxiv: Charting gaps in realistic chart understanding in multimodal llms.
> \[2\] Wiedemer, T., Li, Y., Vicol, P., Gu, S. S., Matarese, N., Swersky, K., ... & Geirhos, R. (2025). Video models are zero-shot learners and reasoners. arXiv preprint arXiv:2509.20328.
> \[3\] Xu, Y., Li, C., Zhou, H., Wan, X., Zhang, C., Korhonen, A., & Vulić, I. (2025). Visual Planning: Let's Think Only with Images. arXiv preprint arXiv:2505.11409.

---

### Official Review · Reviewer_cLBW · 2025-11-01

**Soundness:** 3
**Presentation:** 2
**Contribution:** 3
**Rating:** 4
**Confidence:** 4

**Summary:**

This paper investigates why current vision-language models (VLMs) struggle with understanding data visualizations such as scatter plots. The authors introduce FUGU (Fundamentals of Graph Understanding) — a new diagnostic benchmark designed to systematically test foundational spatial and mathematical reasoning skills necessary for chart interpretation, including counting, locating points, measuring distances, finding extrema, and computing means. Further proposed tasks include correlation, cluster, function, outlier.
Using three representative VLMs (LLaMA-3.2, LLaVA-OneVision, and InternVL3), the study combines behavioral evaluation with activation patching and linear probing to trace information flow through model components.

**Strengths:**

- **Clear motivation and positioning:**
  The paper addresses a relevant and timely question regarding the ability of VLMs to understand charts and data visualizations. The research problem is articulated clearly, focusing on identifying where failures in chart understanding may originate.

- **Well-scoped contributions:**
  The proposed FUGU tasks are thoughtfully designed and cover a useful range of chart-understanding skills (e.g., counting, coordinates, extrema). The use of both causal interventions and linear probes provides an informative way to study model behavior beyond surface-level accuracy.

- **Comprehensive experimental setup:**
  The experiments include several representative models (e.g., LLaMA-3.2, LLaVA-OneVision, InternVL3) and explore different task conditions. The findings offer a plausible explanation that the vision–language interface poses a key challenge, supported by a range of analyses. The visualizations and ablations help illustrate the results.

- **Clarity and reproducibility:**
  The paper is generally well-organized and provides sufficient implementation detail, including appendices that describe task construction and probe configurations. This level of transparency should support reproducibility and future research on multimodal reasoning.

**Weaknesses:**

- **Limited coverage of visualization types:**
  The analysis focuses mainly on Cartesian point-based charts (e.g., line and bar charts), where positional relationships naturally reflect values. However, for non-Cartesian visualizations such as pie charts or radar charts, angular information is equally critical. The current framework does not appear to account for these cases, limiting its generality across broader visualization types.

- **Insufficient consideration of visual encoder scale and capacity:**
  The conclusion that visual encoders preserve nearly 100% of the relevant information is based on a single encoder configuration. The impact of encoder size, architecture, and pre-training strategy on this finding is not examined. Incorporating comparisons across encoder scales would strengthen the validity of this conclusion.

- **Limited mechanistic insight into the vision–language bottleneck:**
  While the work identifies the vision–language interface as the main source of degradation, the analysis does not delve into *why* this interface fails. Potential factors—such as attention bottlenecks, misalignment in cross-modal token fusion, or representational loss during projection—are not explored. A more detailed mechanistic investigation would make the diagnosis more robust.

**Questions:**

- **Nature of the vision–language bottleneck:**
  The paper attributes the core failure to the vision–language interface, but the underlying mechanism remains unclear. Could the authors elaborate on what aspects of attention or projection layers may cause information loss at this stage? Additionally, if information is already lost at this layer, how do the observed performance fluctuations across subsequent layers arise?

- **Encoder size and representation capacity:**
  The results suggest that positional information is fully preserved across tested vision encoders. How sensitive is this finding to encoder scale or architecture? Would smaller or differently pre-trained encoders yield similar preservation patterns?

- **Scope of FUGU task diversity:**
  The benchmark focuses on scatter plots in Cartesian coordinates. Do the authors expect comparable failure behaviors in charts with non-Cartesian or hierarchical structures (e.g., polar plots, radar charts, treemaps)? Clarification on how the diagnostic approach would generalize to such cases would be helpful.

- **Implications for future model design:**
  Since fine-tuning does not appear to resolve the bottleneck, have the authors considered alternative or hybrid architectural directions that could mitigate the issue? Insight into how these findings might inform future multimodal model design would strengthen the discussion.

---

> ### Author Response · Authors · 2025-12-03
> **Response to Reviewer cLBW**
>
> Thank you for your thoughtful comments and questions! Please find our point-by-point responses below:
>
> ## Weaknesses
>
> ### 1: "Limited coverage..."
> Thank you for raising this point. We agree that expanding beyond scatter plots would further strengthen the generality of our framework. We have taken a substantial step in this direction by extending FUGU to include bar charts and line charts. Please see the global response.
>
> ### 2: "Insufficient consideration..."
> We appreciate the opportunity to clarify this point. Our experiments already evaluate three distinct vision encoders that vary meaningfully along the dimensions the reviewer highlights:
> * LLaMA-3.2: CLIP ViT-L/14
> * LLaVA-OneVision: SigLIP-400M
> * InternVL3: InternViT-300M
> These encoders vary in scale, architectural design, and pretraining, and we find that all three exhibit near-perfect linear probe accuracy, supporting our claim that the visual encoding step is not the primary bottleneck.
>
> ### 3: "Limited mechanistic..."
> We appreciate this suggestion and fully agree that deeper mechanistic analysis of why the vision–language interface fails is an exciting direction for future work. Our findings do offer some preliminary clues. First, because the three models we study employ distinct connector designs yet exhibit strikingly similar degradation patterns, the specific form of the connector may matter less than broader factors. Second, fine-tuning partially recovers performance, suggesting that data scarcity or misaligned training objectives contribute to the issue. Finally, our probe results—and concurrent work showing that VLMs must expend many language layers simply to extract basic visual entities—point to a more fundamental challenge in aligning visual and linguistic representation spaces. We view detailed mechanistic investigation of these alignment difficulties as a valuable next step, and we thank the reviewer for highlighting it.
>
> ## Questions
>
> ### Q1
> Thank you for this thoughtful question. We now clarify several hypotheses informed by concurrent studies and our own results.
>
> First, recent work \[1, 2\] shows that VLMs expend many LM layers transforming visual features into representations the language module can use. This aligns with the gradual increase in probe accuracy we observe across LLaMA-3.2’s LM layers, with fluctuations potentially reflecting noise or nonlinear transformations.
>
> Second, we believe that the pretraining objective of the connector, rather than its specific architecture, may be an important limiting factor. Multimodal connectors are typically trained on image-caption alignment, an objective that does not require preserving fine-grained spatial information. As a result, the learned mapping may exclude the structured spatial information our tasks depend on. Our fine-tuning results support this interpretation. Moreover, because the three models we evaluate employ different multimodal interfaces yet exhibit the same degradation at the vision-language handoff, the remaining performance gap is unlikely to arise from a particular connector design. Instead, it may reflect deeper challenges in aligning latent representations for vision and language, consistent with concurrent work \[1, 2\].
>
> ### Q2
> Across all three encoders, we observe uniformly high linear-probe accuracy throughout the vision stack, suggesting the conclusion is robust. A controlled encoder sweep would be valuable future work.
>
> ### Q3
> Thank you for the suggestion. FUGU was designed to be extensible, and we agree that examining non-Cartesian charts is an exciting application. More broadly, our main contribution is the methodological framework—controlled synthetic stimuli combined with causal interventions and probing—rather than a dependency on any specific chart type.
>
> ### Q4
> We thank the reviewer for raising this important point. We have expanded the discussion to highlight several promising directions suggested by our findings:
> * Spatially aware connectors. Image–caption pretraining may learn mappings that discard spatial structure; connectors optimized to preserve spatial relations could mitigate this.
> * Dynamic, prompt-conditioned connectors. Adapting the visual representation to the task described in the prompt may reduce information loss.
> * Reasoning with generative vision models. Instead of forcing all reasoning into the LM, models could reason in the latent space of the vision module, leveraging advances in image/video generation. Recent work shows that video models can solve reasoning tasks zero-shot by generating intermediate frames rather than language tokens \[3, 4\].
>
> ## References
> \[1\] Cohen et al. (2025). Performance gap in entity knowledge extraction across modalities in vision language models.
> \[2\] Fu et al. (2025). Hidden in plain sight: VLMs overlook their visual representations.
> \[3\] Wiedemer et al. (2025). Video models are zero-shot learners and reasoners.
> \[4\] Xu et al. (2025). Visual Planning: Let's Think Only with Images.

---

### Author Response · Authors · 2025-12-03
**Global response**

## Global response
We thank all reviewers for their thoughtful and thorough reviews. Below, we address two common and important threads in the reviews: (1) the limited scope of chart diversity in FUGU, and (2) insufficient discussion of the model limitations we identified.

## Limited scope of chart diversity in FUGU

Multiple reviewers raised the concern that our dataset is limited to scatter plots, and therefore the generalizability of our findings to other chart types is unclear. We agree with the reviewers that the work would be strengthened by including other plot types—we have therefore taken a significant step in this direction by **extending FUGU to include two additional chart types: bar charts and line charts**.

**We reproduced our analyses from sections 4.1, 4.3, 4.4, and 4.5 on bar and line charts** (behavioral evaluation, point listing accuracy, chain-of-thought interventions, and linear probes) for our three main models (LLaMA-3.2, LLaVA-OneVision, InternVL3). We found that **the key patterns observed in scatter plots are largely recapitulated** in these new settings, reinforcing our original conclusions:

* **Behavioral evaluation** (Section 4.1). Model performance degrades as the number of bars/data points in the plot increases.
* **Point listing accuracy** (Section 4.3). Models often list the values of bars or data points in their chain-of-thought before performing arithmetic operations on these values; model accuracy in value extraction worsens for more complex plots containing more data points, and these errors propagate to downstream reasoning.
* **Chain-of-thought interventions** (Section 4.4). Providing models with ground truth lists of values generally improves performance across tasks for bar and line charts, indicating that accurate value/coordinate extraction is indeed an important bottleneck.
* **Linear probes** (Section 4.5). Value/coordinate information is linearly decodable from vision layers with nearly perfect accuracy; this information then gets distorted after the vision-language handoff, indicated by a significant drop in probe test accuracy on language layers.

Full results as well as details about bar and line chart generation are provided in a new appendix section in the revised PDF (Appendix H), and the code for generating all chart types will be made publicly available.

## Insufficient discussion of model limitations

Reviewers raised two concerns regarding our discussion of the nature of the model limitations we identified: (a) insufficient discussion of avenues for model improvements in our discussion, and (b) whether the coordinate extraction bottleneck represents a fixed architectural limitation. We have expanded the discussion in the revised PDF to address these points in detail, and we summarize the concerns and our responses below.

**Insufficient discussion of avenues for model improvements**. Several reviewers suggested that we expand our discussion to include potential avenues for improvements to VLMs based on our results. There is an abundance of model improvement ideas that are well motivated by our results, and we have added a paragraph to the discussion detailing three of them: (1) connectors developed specifically for spatial reasoning tasks, (2) dynamic, prompt-conditioned connectors that selectively attend to visual information depending on the task described in the prompt, and (3) reasoning with generative vision models.

**Fixed architectural limitations**. Some reviewers were concerned that we overstate our findings as fundamental architectural limits of VLMs. We appreciate the opportunity to clarify this point: we do not argue that the vision-language interface imposes a fixed, immutable architectural ceiling. Rather, across models and tasks, we consistently observe a recurrent point of degradation: task-relevant spatial information is well-represented in the vision encoder but is not reliably accessed or used by the language module. We call this a “bottleneck” to describe a systematic information-flow inefficiency—not to assert that it is the only source of failure or that it cannot be mitigated in existing architectures. Our results (including LM-specific errors and fine-tuning gains) explicitly support this more scoped interpretation.

---

### Meta-Review · Area_Chair_Y7Mp · 2025-12-31

**Summary:**

The papers conducts a study on the capabilities of Visual Language Models (VLMs) at basic data visualization tasks. It proposes FUGU a framework based on the analysis of scatter plots where the performance of models, via linear probes and activation patching, is analyzed under task of different levels of complexity. The key finding made by the paper is that even models with powerful vision encoders fail to properly pass information to the language model part, which seems to be the “bottleneck” where lost of performance originates.

The reviewers found the content of the paper clear and well-structured, the problem investigated therein interesting, and the proposed FUGU framework flexible enough to facilitate analysis.

On the original rebuttal, concerns regarding the limited coverage of visualization/chart  types was addressed to some degree. Likewise, the rebuttal clarified that a variety of visual encoders with different scales and capacity were already included in the evaluation.

On the downside, several major concerns were not fully addressed by the initial rebuttal, namely, proposing/testing solutions for the identified problem, the generality of the reported observations to all visualization/charts types and to tasks that do not rely on the extraction of (x,y) coordinates, and the currently-unreported user study, which leave the manuscript in a rather borderline position.

Considering the potential that investigating/solving the problem at hand  could have, I would encourage the authors to refrain from getting the paper accepted/published in its current form and aim at submitting a stronger version based on the content already provided in the rebuttal plus some additional content addressing the outstanding concerns listed above. I believe an improved version of the manuscript is likely to have stronger impact.

**Reviewer Concerns:**

Addressed Concerns:

- Reviewer cLBW

    - Limited coverage of visualization chart types.

    - Insufficient evaluation of visual encoders with different characteristics, i.e. scale and capacity.

- Reviewer jg4B:

    - Limited coverage of visualization chart types.

- Reviewer 4DYu

    - Limited coverage of visualization chart types.

- Reviewer oyTB

    - Limited coverage of visualization chart types and complexity.

    - Finetuning results do not fully support the claims made.


Outstanding Concerns:


- “Reviewer cLBW”

    - Limited mechanistic insight into the problem. As accurately indicated by the reviewer, the paper limits itself to highlighting an existing issue, no solution is proposed/tested to alleviate the effect of the detected issue.

- Reviewer jg4B

    - Not possible to assess whether the made observations and reached conclusions would hold across all visualization/chart types.

    - The paper's contribution is primarily diagnostic. This is accurate, no solution for the problem is proposed/tested .

    - No human-based evaluation, e.g. a user study, is conducted. The authors claim they conducted such evaluation, which aligned with their findings, but decide to not include it in the manuscript. This is somewhat suspicious an raises questions. Beyond that, no details on the conducted user study, which could have given some credibility to the statement, were reported in the rebuttal.

    - Tasks within the proposed FUGU framework seem to rely on the accurate extraction of (x,y) coordinates which would bias the aspect being investigated by the proposed general framework.

- Reviewer 4DYu

    - All the concerns (1) were addressed

- Reviewer oyTB

    - Limited insight on the nature of the identified “bottleneck” which seems to be task dependent.

**Reviewer Scores:**

This paper is very borderline, while the authors made an effort to properly address some of the concerns raised by the reviewers, there are some aspects that were still open/inconclusive after the initial rebuttal. There are some concerns that might require a significant modification, e.g. proposed/tested solutions for the identified problems, that might not had been feasible during the discussion period. I believe this manuscript might not had being able to leave that boundary state after the discussion period.

---

### Decision · Program_Chairs · 2026-01-26

Reject